# Elucidating relationships between *P. falciparum* prevalence and measures of genetic diversity with a combined genetic-epidemiological model of malaria

**Jason A. Hendry** [1] *, **Dominic Kwiatkowski** [1,2,3,4], **Gil McVean** [1,3]

**1** Big Data Institute, Li Ka Shing Centre for Health Information and Discovery, University of Oxford, Oxford, United Kingdom, **2** Wellcome Centre for Human Genetics, University of Oxford, Oxford, United Kingdom, **3** Medical Research Council Centre for Genomics and Global Health, University of Oxford, Oxford, United Kingdom, **4** Wellcome Sanger Institute, Cambridge, United Kingdom

* jason.hendry@well.ox.ac.uk

**Data Availability Statement:** All relevant data are within the manuscript and its Supporting information files. All of the code developed as part

## Abstract

There is an abundance of malaria genetic data being collected from the field, yet using these data to understand the drivers of regional epidemiology remains a challenge. A key issue is the lack of models that relate parasite genetic diversity to epidemiological parameters. Classical models in population genetics characterize changes in genetic diversity in relation to demographic parameters, but fail to account for the unique features of the malaria life cycle. In contrast, epidemiological models, such as the Ross-Macdonald model, capture malaria transmission dynamics but do not consider genetics. Here, we have developed an integrated model encompassing both parasite evolution and regional epidemiology. We achieve this by combining the Ross-Macdonald model with an intra-host continuous-time Moran model, thus explicitly representing the evolution of individual parasite genomes in a traditional epidemiological framework. Implemented as a stochastic simulation, we use the model to explore relationships between measures of parasite genetic diversity and parasite prevalence, a widely-used metric of transmission intensity. First, we explore how varying parasite prevalence influences genetic diversity at equilibrium. We find that multiple genetic diversity statistics are correlated with prevalence, but the strength of the relationships depends on whether variation in prevalence is driven by host- or vector-related factors. Next, we assess the responsiveness of a variety of statistics to malaria control interventions, finding that those related to mixed infections respond quickly ($\sim$ months) whereas other statistics, such as nucleotide diversity, may take decades to respond. These findings provide insights into the opportunities and challenges associated with using genetic data to monitor malaria epidemiology.

of this manuscript, including forward-dream, is available on GitHub (https://github.com/JasonAHendry/fwd-dream).

**Funding:** This study was supported by the Wellcome Trust https://wellcome.ac.uk/ (206194, 090770, 204911, 100956/Z/13/Z to GM, and 109107/Z/15/Z to JH) and the Li Ka Shing Foundation https://www.lksf.org/ (to GM). The computational aspects of this research were supported by the Wellcome Trust Core Award Grant Number 203141/Z/16/Z and the NIHR Oxford BRC. The views expressed are those of the author (s) and not necessarily those of the NHS, the NIHR or the Department of Health. The funders had no role in study design, data collection and analysis, decision to publish, or preparation of the manuscript.

**Competing interests:** The authors have declared that no competing interests exist.

## Author Summary

Knowledge of how the prevalence of *P.falciparum* malaria varies, either between regions or through time, is critical to the operation of malaria control programs. Yet obtaining this information through traditional methods presents many challenges. Parasite genetic data is increasingly accessible, and may provide an alternative means to estimate *P.falciparum* prevalence in the field. However, our understanding of how the genetic diversity of parasite populations relates to prevalence is limited, and suitable models to guide our understanding are largely lacking. Here, we merge two classical models—the Ross-Macondald and the Moran—to produce a framework in which the relationships between parasite genetic diversity and prevalence can be explored. We find that several genetic diversity statistics are correlated with prevalence, although to differing degrees, and over different time scales. Overall, statistics related to mixed infection are robustly and rapidly responsive to changes in prevalence, suggesting they may be a useful focal point for the development of malaria surveillance methods that harness genetic data.

## Introduction

It is widely accepted that relationships exist between the regional epidemiology of malaria and the genetic diversity of local parasite populations. For example, the origin and spread of anti-malarial drug resistance, the rate and directions of parasite migration, and the local intensity of transmission all have an impact on population genetic diversity (reviewed in [1–7]). In most cases, however, the precise nature of these relationships remains unclear. From a modelling perspective, exploring these relationships would require that both genetic processes (including mutation, drift and meiosis) and epidemiological ones (including the transmission dynamics and life cycle of malaria) are combined into a single framework. At present such integrated models are rare, yet without them, parasite genetic data will be under-utilized as a resource for malaria surveillance.

One epidemiological parameter of central importance to malaria surveillance is transmission intensity, as it is used by National Malaria Control Programs (NMCPs) to prescribe malaria control interventions and assess their efficacy [8]. NMCPs can attempt to measure transmission intensity with a variety of statistics. These include the basic reproduction number ($R_0$), defined as the number of secondary infections produced by a single infected individual in a naive population; the entomological inoculation rate (*EIR*), defined as the mean number of infectious bites received by an individual per annum; parasite prevalence or rate (*PR* or *PfPR* if the focus is *P. falciparum*), defined as the fraction of individuals in the population carrying detectable infection; or simply the rate of clinical incidence (reviewed in [9]). However, there are well-documented issues with all of these statistics. Though a theoretical gold-standard, $R_0$ is difficult to measure in practice, with estimation methods relying either on exploiting equilibrium relationships with other measures of transmission, or formulae involving several poorly-characterised parameters [9, 10]. The *EIR* suffers from small-scale variability in mosquito density, a lack of standardisation across mosquito catching methods, and difficulties associated with catching sufficient mosquitoes when transmission intensities are low [9, 11, 12]. Rates of clinical incidence are confounded by variation in acquired immunity and treatment seeking behaviour, as well as incomplete record keeping [12]. Parasite prevalence is the most widely collected measure and has been used as the basis of large-scale maps [13–15], yet it requires prohibitively extensive sampling at low transmission intensities [12], and must address biases in detection power that may arise from infection-course and age-dependent variation in

parasitemia [9, 16]. Thus, a means to either estimate or improve existing estimates of transmission intensity with genetic data would be valuable.

The problem of estimating an epidemiological parameter like transmission intensity from genetic data is superficially similar to the demographic inference problems that are commonly encountered in population genetics. For example, a multitude of methods now exist for estimating effective population size ($N_e$) from genetic data [17–20], and it could be hypothesized that the regional transmission intensity of malaria is a function of the $N_e$ of the local parasite population. However, there are at least two challenges unique to epidemiological inference from malaria genetic data that make it a distinctive and more difficult problem.

First, it is not clear that the classical models in population genetics (including the Wright-Fisher, Moran, and other models that converge to Kingman's $n-$Coalescent in the infinite population limit [21]) that are often employed by demographic inference methods are suitable for malaria. The life cycle of *P. falciparum* involves oscillating between human host and mosquito vector populations, which may be of different sizes, and may also induce different rates of drift and mutation. Within both the host and the vector, parasite populations likely experience bottlenecks (for example, as the ookinetes penetrate the midgut wall of the vector), exponential growth phases (merozoites replicating in the blood of the host), and interactions with the host or vector immune system. Indeed, there has been work demonstrating that the malaria life cycle simultaneously intensifies drift and selection; a result contrary to what is expected under a Wright-Fisher model [22]. So, while classical models in population genetics have the advantage of being extensively studied and mathematically tractable, with a known set of relationships between equilibrium genetic diversity statistics and demographic parameters, they do not readily apply to *P. falciparum*. Finally, even if these models did apply, the relationships between demographic parameters (such as $N_e$) and epidemiological ones (such as transmission intensity) would need to be defined.

Conversely, the epidemiological models that have been designed to reflect malaria biology and transmission do not explicitly incorporate genetic processes. The most well known class of epidemiological models are the compartment-based models, where individual hosts and vectors transition between compartments which can represent a variety of disease states (such as susceptible, infected, or immune), and the overall population is represented by the total number of hosts and vectors occupying each state (reviewed in [23, 24]). A canonical compartment model for malaria is the Ross-Macdonald, where hosts and vectors can be either susceptible or infected [24]. Conveniently, in these models, the equilibrium prevalence of infected hosts and vectors is a function of the parameters specifying the transition rates between compartments. However, the absence of genetic processes (and indeed, individual parasites) means they offer no insight into how parasite genetic diversity relates to these transition rates or, as a corollary, any epidemiological parameters derived from them. Thus, at least with respect to malaria, the traditional modelling landscape cannot address questions that involve both genetic data and epidemiology.

A second challenge specific to epidemiological inference using genetic data is that the ultimate aim is often to inform disease control and, as a result, the time-dimension of the inference is of critical importance. Many demographic inference methods base their estimates of $N_e$ on distributions of coalescent times between segments of DNA. As these coalescent events typically occur on the scale of $N_e$ generations in past, the estimates are historical; reflecting the average population size over hundreds or thousands of generations. Such approaches are not suitable for disease control, where policy decisions need to be made on the basis of information about the near-present, or predictions about the future.

In view of these challenges, the development of integrated genetic-epidemiological models is a critical undertaking for the future of malaria genomic surveillance. To date, there have been two notable examples of such integrated models. The first, developed by Daniels et al. in 2015, was designed to support epidemiological inference from single-nucleotide polymorphism (SNP) data, collected during a period of intensified intervention in Thiès, Senegal [25]. Here, the model was used with an Approximate Bayesian Computation (ABC) algorithm to independently corroborate a decline and rebound in transmission intensity on the basis of 24-SNP barcodes [25]. More recently, Watson et al. have developed a model of substantially greater epidemiological complexity—including features such as individual vectors and a vector development cycle, six host infection states, host age and immunity—and used this model to estimate parasite prevalence in five locations from Uganda and Kenya after fitting a single parameter in the model to SNP data [26]. In both these cases, the focus on SNP barcodes reflected the data available. Yet, as *P. falciparum* whole-genome sequencing (WGS) data continues to increase [27], there is a need for modelling frameworks that can investigate the broader suite of genetic diversity statistics calculable from WGS data.

Here, we aim to address the lack of integrative genetic-epidemiological models by developing a new forward-time model called `forward-dream`. `forward-dream` merges the Ross-Macdonald model with a continuous-time intra-host and intra-vector Moran model, and further incorporates meiosis within the vector (allowing for multiple oocysts), multiple infection (by either super- or co-infection), and a representation of the transmission bottlenecks. Implemented as a stochastic simulation, we use the model to explore relationships between measures of genetic diversity and parasite prevalence, both at equilibrium and in response to malaria control interventions that perturb equilibrium. We confirm that a variety of genetic diversity statistics are correlated with parasite prevalence, although to varying degrees and over different time-scales. In addition, we find that interventions that affect the duration of infection in hosts have a greater influence on parasite genetic diversity than those that influence vector biting rate or density. Overall, our results suggest that statistics based on the complexity of infection (COI) are strongly, robustly, and rapidly responsive to changes in prevalence, highlighting their potential value for malaria surveillance.

## Results

### Developing a model of *P. falciparum* malaria transmission and evolution

We developed an agent-based simulation of *P. falciparum* malaria incorporating features of its transmission and life cycle, as well as explicitly modelling the genetic material of parasites. Our integrated genetic-epidemiological model is called `forward-dream` (**forward**-time **d**rift, **r**ecolonisation, **e**xtinction, **a**dmixture and **m**eiosis) and is comprised of three layers: (1) a stochastic epidemiological layer, which controls how malaria spreads through a population of hosts and vectors and reaches equilibrium; (2) a stochastic infection layer, which controls the behaviour of malaria parasites during individual transmission events and within individual hosts and vectors; and (3) a stochastic genetic layer, which controls how the genetic material of individual parasites is represented, mutated and recombined (Fig 1). We describe each layer below and provide additional information, including a discussion of parameterisation, in the S1 Appendix.

**Epidemiological layer.**   For the epidemiological layer we implemented a stochastic, agent-based version of the Ross-Macdonald model (reviewed in [24]). In this model, a fixed number of hosts ($N_h$) and vectors ($N_v$) alternate between susceptible and infected based on four fixed

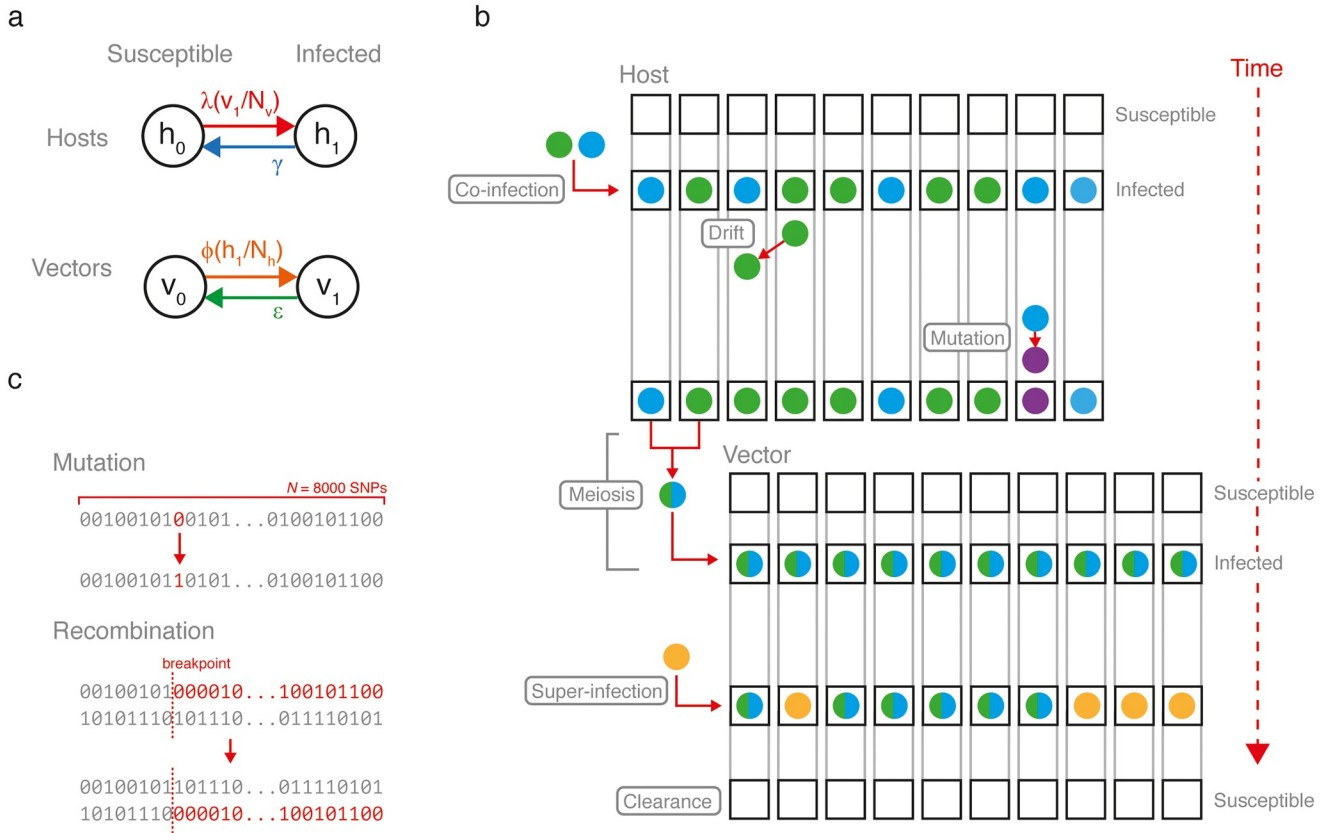

**Fig 1. Schematic of `forward-dream`.** (a) The epidemiological layer. Hosts and vectors oscillate between susceptible and infected compartments according to a Ross-Macdonald model. (b) The infection layer. The capacity of individual hosts and vectors to be infected is represented by a fixed number of sub-compartments (black boxes) each which can harbour a unique parasite genome (colored circles). In a susceptible host/vector, all sub-compartments are empty. Upon infection, all sub-compartments are populated. Drift and mutation occur among sub-compartments according to a continuous-time Moran model. Parasite genomes undergo meiosis during transmission from host to vector. Super-infection can occur, resulting in an average of half of all sub-compartments being replaced with newly transmitted parasite genomes. Note that the infection layer can be nested within the Ross-Macdonald model. (c) The genetic layer. The genome is represented by a fixed-length array of 0's and 1's. Mutation is reversible, converting 0 to 1 or 1 to 0. Recombination occurs during meiosis.

rate parameters (Fig 1a). The model can be described by two coupled differential equations:

$$\frac{dh_1}{dt} = h_0(v_1/N_v)\lambda - h_1\gamma,$$

$$\frac{dv_1}{dt} = v_0(h_1/N_h)\phi - v_1\epsilon,$$

where $h_0$ and $h_1$ are the number of susceptible and infected hosts, respectively, with $N_h = h_0 + h_1$; and $v_0$ and $v_1$ are the number of susceptible and infected vectors, respectively, with $N_v = v_0 + v_1$. Note that in this model $\lambda$ and $\phi$ are compound parameters representing the rate at which hosts and vectors become infected. In particular,

$$\lambda = b(N_v/N_h)\pi_h,$$

where $b$ is the daily vector biting rate, $(N_v/N_h)$ gives the vector density, and $\pi_h$ is the probability that an infectious bite from a vector produces an infected host (the vector-to-host transmission

efficiency). Similarly, we have,

$$\phi = b\pi_v,$$

where $b$ is again the daily vector biting rate and $\pi_v$ is the probability that a vector that bites an infected host becomes infected (the host-to-vector transmission efficiency). We allow for mixed infections however, for the epidemiological layer, they have no consequence: both clonal and mixed infections are assigned to the infected compartments ($h_1$ or $v_1$).

Note that the epidemiological layer dictates the equilibrium prevalence of infection in hosts ($X_h = h_1/N_h$) and vectors ($X_v = v_1/N_v$). In particular, the equilibrium prevalence in hosts is given by:

$$X_h = \frac{\lambda\phi - \gamma\epsilon}{\lambda\phi + \phi\gamma} \tag{1}$$

and in vectors by:

$$X_v = \frac{\lambda\phi - \gamma\epsilon}{\lambda\phi + \lambda\epsilon}. \tag{2}$$

We have confirmed that our implementation of the Ross-Macdonald model in `forward-dream` converges to the expected equilibrium prevalence values (Fig 2a and S1 Fig). In total, the behaviour of the epidemiological layer is specified by seven parameters.

**Infection layer.** The infection layer specifies the biology of individual infection and transmission events. This includes a representation of: (i) the capacity of hosts and vectors to harbour infection, (ii) the evolution of infections within hosts and vectors, and (iii) the transmission of infection between hosts and vectors.

We model an individual host's capacity to be infected with $n_h$ within-host sub-compartments (Fig 1b). When an individual host is in the susceptible state (i.e. is uninfected), all $n_h$ sub-compartments are empty. When an individual host becomes infected, all $n_h$ sub-compartments simultaneously become populated. Each sub-compartment can potentially harbour a unique parasite genome (such that the maximum complexity of infection $k = n_h$), although typically multiple sub-compartments will be occupied by identical, or near-identical, genomes. For example, if a host is infected by a vector carrying a single distinct parasite genome, all $n_h$ sub-compartments will initially be occupied by that genome. Alternatively, if a host is co-infected by a vector carrying two distinct parasite genomes, its sub-compartments will be drawn from a mixture of the two genomes.

Moving forward in time, the *P. falciparum* infection of a host evolves according to a continuous-time Moran process for a population with $n_h$ individuals, parameterised by a drift rate ($d_h$) and mutation rate ($\theta_h$) [28]. We have confirmed that our implementation of the Moran process yields fixation times consistent with theoretical expectation (S2 Fig). The Moran process continues until the infection is cleared and the host returns to the susceptible state, with all sub-compartments becoming simultaneously empty. For hosts, clearance occurs at rate $\gamma$, as specified in the epidemiological layer. The infection of a vector evolves according to a comparable process as in the host, but with a set of parameters $n_v$, $d_v$, $\theta_v$, and $\epsilon$.

In `forward-dream`, a *P. falciparum* infection is transmitted from a vector to a host through a transmission bottleneck, such that not all of the parasites within the infectious vector establish themselves in the host. In particular, a random subset of all parasites $\bar{v} \subseteq v$ (where $v = \{v_1, v_2, ..., v_{n_v}\}$ is the set of all parasites in the vector) is transmitted. The number of

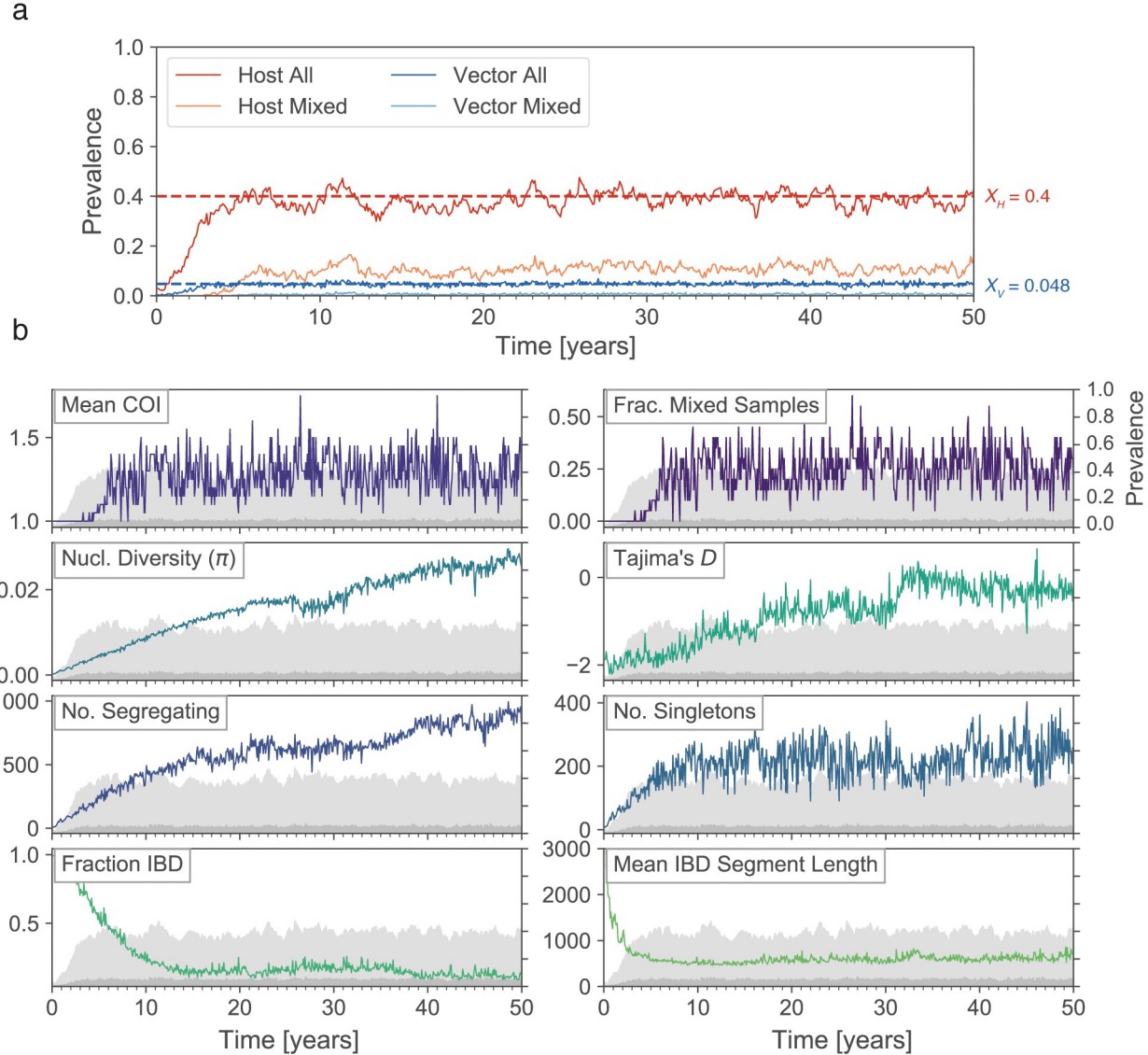

**Fig 2. Simultaneously monitoring parasite prevalence and genetic diversity using `forward-dream`.** (a) A `forward-dream` simulation seeded with ten infected hosts harbouring identical parasite genomes and run for 50 years. Prevalence of all infected hosts ('Host All', which corresponds to *PfPR*) and multiply-infected hosts ('Host Mixed') is indicated by red and pink lines, respectively. The same is shown for vectors in blues. The prevalence of infected hosts and vectors fluctuates around their Ross-Macdonald equilibrium values ($X_h$ and $X_v$), indicated with red and blue horizontal lines, respectively. (b) The same simulation as in (a), but visualizing eight genetic diversity statistics that were computed by collecting parasite genomes from twenty randomly selected infected hosts every 30 days. The statistics are defined in S1 Table. For reference, the light and dark grey shaded areas show the host and vector prevalence (right y-axis), corresponding to the red and dark blue lines in panel (a).

transmitted parasites $n = |\bar{v}|$ is drawn from a truncated binomial:

$$n \sim max[1, Bin(n_v, p_v)].$$

This results in an average of $n_v p_v + (1 - p_v)^{n_v}$ parasites passing through the transmission bottleneck; the size of the bottleneck is controlled by $p_v$. For all of the simulations presented here, $p_v = 0.2$ and $n_v = 20$, resulting in an average of $\simeq 4$ parasites passing through the bottleneck.

Note that in cases where $n > 1$ and $\bar{v}$ contains unique parasites, co-infection may occur. If the vector is infecting a susceptible (i.e. uninfected) host, $n_h$ parasites are drawn with

replacement from $\bar{v}$ with each having an equal probability $(1/n)$ of being drawn. These $n_h$ parasites then populate the $n_h$ host sub-compartments. Alternatively, super-infection occurs if the host is already infected. Defining the parasites already within the host by the set $h = \{\eta_1, \eta_2, ..., \eta_{n_h}\}$, we first create the union $h \cup \bar{v}$. From this set $n_h$ parasites are drawn, where each parasite has probability $1/2n_h$ or $1/2n$ of being drawn, if it is from $h$ or $\bar{v}$, respectively. As a consequence, on average super-infection results in half of the within-host compartments being occupied by new parasites.

The transmission from host to vector is the same as above, but with the addition of meiosis. In brief, the $n$ parasite strains selected at random from the infected host may undergo meiosis before populating the the $n_v$ within-vector sub-compartments. The meiosis model is based on a simplified implementation of our previously published meiosis simulator, `pf-meiosis` [29]. The number of recombination events is drawn from a Poisson distribution scaled with respect to chromosome length, such that an average of one cross-over event occurs per bivalent during meiosis. Recombination breakpoints are sampled uniformly from along the chromosome. The model includes multiple oocysts, allowing for parallel rounds of meiosis to occur during a single transmission event, with the number of oocysts being drawn from a truncated geometric distribution: $\sim min[10, Geo(p_{oocysts})]$.

In total, the infection layer is specified by nine parameters.

**Genetic layer.**    The genetic layer of `forward-dream` describes the malaria genome model. We represent the genetic material of an individual parasite as a single fixed-length array of zeros and ones, defined by the parameter $N_{snps}$ (Fig 1c). In effect, this array represents a single chromosome marked with $N_{snps}$ single-nucleotide polymorphisms (SNPs). Mutation is symmetric and reversible. The only parameter specific to the genome evolution layer is $N_{snps}$.

Overall, `forward-dream` is specified by 17 parameters (Table 1). It is implemented in Python and available on GitHub at https://github.com/JasonAHendry/fwd-dream.

**Table 1. Complete list of simulation parameters for `forward-dream`.** Values given represent those of a simulation with a host ($X_h$) and vector ($X_v$) prevalence of 0.65 and 0.075, respectively. This corresponds to the *Initialise* epoch of all malaria control intervention simulations (see below). For details on how parameter values were selected see the S1 Appendix.

| Parameter | Definition | Value |
|---|---|---|
| $N_h$ | Number of hosts | 1000 |
| $N_v$ | Number of vectors | 5000 |
| $b$ | Biting rate (per vector per day) | 0.25 |
| $\pi_h$ | Vector-to-host transmission efficiency | 0.1 |
| $\pi_v$ | Host-to-vector transmission efficiency | 0.1 |
| $\gamma$ | Host infection clearance rate | 0.005 |
| $\epsilon$ | Vector infection clearance rate | 0.2 |
| $n_h$ | Maximum COI for hosts | 20 |
| $n_v$ | Maximum COI for vectors | 20 |
| $d_h$ | Drift rate in hosts (events per day per host) | 2 |
| $d_v$ | Drift rate in vectors (events per day per vector) | 2 |
| $\theta_h$ | Probability of mutation event given drift has occurred in hosts | $1.25 \cdot 10^{-4}$ |
| $\theta_v$ | Probability of mutation event given drift has occurred in vectors | $1.25 \cdot 10^{-4}$ |
| $p_h$ | Probability a given parasite genome passes through host-to-vector transmission bottleneck | 0.2 |
| $p_v$ | Probability a given parasite genome passes through vector-to-host transmission bottleneck | 0.2 |
| $p_{oocysts}$ | Number of oocysts drawn from $\sim Geo(p_{oocysts})$ | 0.5 |
| $N_{snps}$ | Number of SNPs modelled per parasite genome | 8000 |

## Relationships between parasite prevalence and genetic diversity at equilibrium

Within the `forward-dream` framework, it is straightforward to monitor the fraction of hosts that are infected ($h_1/N_h$), which corresponds to the most ubiquitously collected measure of transmission intensity, parasite prevalence (*PfPR*) (Fig 2a). It is also possible to monitor the fraction of vectors infected ($v_1/N_v$), which corresponds closely to the sporozoite rate (*SP*). Finally, the genetic diversity of the parasite population can be monitored by collecting parasite genomes from a sample of infected hosts and simulating DNA sequencing (see Materials and methods) (Fig 2b).

We sought to use `forward-dream` to elucidate relationships between *PfPR* and the genetic diversity of the parasite population. To this end, we varied *PfPR* across simulations and observed the resulting differences in parasite genetic diversity. Within the Ross-Macdonald framework, *PfPR* is a function of the four rate parameters (see Eq 1). In nature, what underlies prevalence differences observed between two geographies or points in time is often unknown, and likely the outcome of a myriad of epidemiological and environmental factors. To achieve different *PfPR* values in `forward-dream`, we choose three parameters to vary separately: (i) the human infection clearance rate ($\gamma$), which may vary between sites if, at one site there is quicker recourse to treatment, or differing proportions of symptomatic and asymptomatic individuals; (ii) the vector biting rate ($\lambda$), which may vary as a consequence of differences in vector species, environmental conditions, or the presence of bednets; and (iii) the number of vectors ($N_v$)—which is influenced by climate and weather, local geography, and also by insecticide-based interventions (see [30]). We tuned each of these parameters to achieve equilibrium *PfPR* values that varied from 0.1 to 0.8 in a population of one thousand human hosts (S3 Fig). At each prevalence value, `forward-dream` was seeded with forty infected hosts carrying identical parasites and then run to equilibrium. After reaching equilibrium, simulations were continued for an additional 10 years, during which time parasite genomes were collected every thirty days by sampling from the infected host population. Since we also wanted to explore the noise distributions of different genetic diversity statistics and how they are influenced by sample size, at each sampling time we collected parasite genomes from five randomly-drawn samples ranging in size from 20 to 100 infected hosts.

The genetic diversity statistics calculated from these parasite genomes are described in S1 Table. The statistics can be divided into three broad categories: (i) those related to mixed infections, which includes the fraction of mixed samples and the mean complexity of infection (COI); (ii) those that are related to the size and shape of the genetic genealogy of the sample, which includes the number of segregating sites, the number of singletons, nucleotide diversity ($\pi$), Watterson's Theta ($\theta_w$), and Tajima's *D*; and (iii) those that summarize the structure of identity-by-state (IBS) and identity-by-descent (IBD) within the population, including, between a pair of samples, the average fraction of the genome in IBD or IBS and the average length of an IBD or IBS segment. We note that these statistics are not independent, indeed many are co-linear (S4 Fig), however they reflect commonly-used measures of genetic diversity in population genetics.

We conducted a total of thirty replicate simulations at each prevalence level and aggregated all the samples of a given size to explore both relationships with *PfPR* and sampling noise. The relationships for all eleven genetic diversity statistics are shown (Fig 3a and S5–S7 Figs), partitioned by which epidemiological parameter was varied. Linear regression was used to determine the fraction of the variance ($r^2$) in each genetic diversity statistic that could be explained by variation by *PfPR*, and how this changed with sample size (Fig 3b). We observed that nearly all genetic diversity statistics have a considerable proportion of their variance explained by

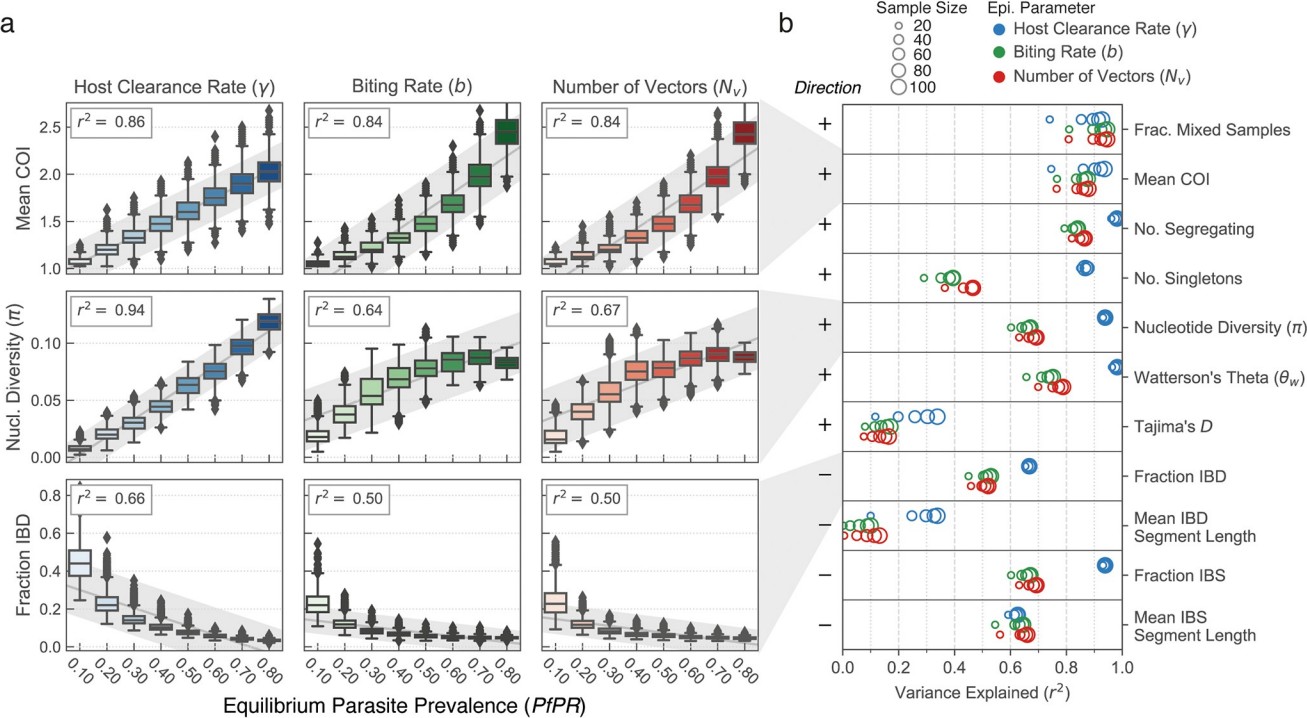

**Fig 3. Equilibrium relationships between genetic diversity and parasite prevalence.** (a) Boxplots showing the equilibrium parasite prevalence in hosts (x-axis) versus genetic diversity of the parasite population, by different statistics. Each individual box summarizes 30 replicate simulations at the indicated prevalence; in each, 40 infected hosts were sampled every 30 days for ten years to collect parasite genomes from which diversity statistics were computed. Left, middle, and right columns show relationships when equilibrium parasite prevalence is varied as a function of the host clearance rate ($\gamma$, in blue), vector biting rate ($b$, in green) or number of vectors ($N_v$, in red). The variance explained in an ordinary linear regression ($r^2$) is shown at the top left for each relationship, and in (b) the variance explained is shown across a larger panel of genetic diversity statistics and as a function of sample size. The variation in $r^2$ as a function of sample size is indicated by point size. Slope of the line of best fit is indicated at left (+, increasing; -, decreasing). Boxplots for all statistics in (b) can be found in S5–S7 Figs.

prevalence. For many statistics—the fraction of mixed samples, mean COI, number of segregating sites, nucleotide diversity, Watterson's $\theta$, and fraction IBS—the $r^2$ exceeded 0.9 for at least one epidemiological driver. The two statistics with the lowest $r^2$ values were Tajima's $D$ and the mean IBD segment length. A stable value for Tajima's $D$ would suggest that the parasite population's genealogical tree has a consistent shape across different prevalence values. One possible cause for the modest $r^2$ values we observe for Tajima's $D$ is recurrent mutation that results from a finite genome size. A lower $r^2$ for the mean IBD segment length may be in part related to a lack of power to detect small IBD segments at higher prevalence values.

A second major observation is that there are pronounced differences in the variance explained by *PfPR* when different epidemiological parameters drive variation in prevalence. In particular, varying host clearance rate ($\gamma$) results in a significantly higher $r^2$ for all of the genealogy statistics, as well as for all of the IBD and IBS statistics, excluding mean IBS segment length. For example, the variance in nucleotide diversity explained by *PfPR* drops from 94% to 60% (for samples of size 40) when parasite prevalence is modulated by the number of vectors ($N_v$), instead of the host clearance rate ($\gamma$). This reduction in explanatory power is associated with a plateau in diversity at higher prevalence values observed when either the number of vectors ($N_v$) or biting rate ($b$) are increased. In contrast, the two statistics related to mixed infections have consistently high $r^2$ values (>75%), regardless of which epidemiological parameter is driving prevalence variation.

The $r^2$ value is a reflection of the signal-to-noise ratio for different genetic diversity statistics, and may be influenced by sample size. Here, we find that for some statistics there is considerable benefit to collecting additional samples, whereas for others there is only little benefit (Fig 3b and S8 Fig). When prevalence change is driven by the number of vectors ($N_v$) or biting rate ($b$), the variance in the fraction of mixed samples explained by $PfPR$ climbs from 0.81 to 0.94 as the sample size is increased from 20 to 100. For the same increase in sample size, the $r^2$ of nucleotide diversity increases only from 0.62 to 0.68. The $r^2$ of the mean IBD segment length also increases substantially with an increase in the number of samples collected (from 0.01 to 0.15). Here, a small sample size can result in a very high variance in mean estimates, due to the chance detection of rare, long IBD segments. We found that increasing from 20 to 100 samples reduced the variance in mean IBD segment length by up to 60% (S8 Fig).

There are multiple sources for the noise that remains even after increasing sample size, related to the stochastic nature of both the epidemiological and genetic processes. Epidemiological noise means that prevalence fluctuates around its equilibrium value, and this can impact some diversity statistics. Noise on the genetic level can exist deep within the structure of the genealogical tree (nearer to the root), but also also more recently. To assess these contributions, we performed a variance decomposition analysis, determining what fraction of the variation in different genetic diversity statistics was occurring within- versus between- replicate simulations (S9 Fig). Since we collect parasite genetic data for 10 years for each replicate simulation, variation on a timescale longer than this will manifest as between- rather than within-simulation. Overall, we found that the majority of variation occurred between replicate simulations for the genealogy related statistics, indicating that they vary over long timescales (S9 Fig). Conversely, more than 90% of the variation in COI statistics existed within individual simulations, indicative of a short timescale of variation.

## Non-equilibrium relationships between parasite prevalence and genetic diversity

An important application of our model is in settings where malaria control interventions are actively occurring. In such cases, and also in cases with seasonal variation in parasite prevalence, it is unlikely that the parasite population will be in equilibrium. Thus, our second aim with `forward-dream` was to explore which measures of genetic variation are most predictive of instantaneous $PfPR$ in non-equilibrium settings.

**Malaria control interventions.**   In order to understand how genetic diversity statistics relate to $PfPR$ in scenarios where a malaria control intervention has been deployed, we developed a framework where individual `forward-dream` simulations pass through three distinct epochs: *Initialise*, *Crash* and *Recovery* (Fig 4). In the *Initialise* epoch, simulations are run until equilibrium at a parasite prevalence of 0.65, under the parameters listed in Table 1. At the start of the *Crash* epoch, one of either the host clearance rate ($\gamma$), the vector biting rate ($b$), or the number of vectors ($N_v$), is changed such that the new equilibrium $PfPR$ is 0.2. The parameter change occurs incrementally over a period of thirty days following a logistic transition function, as to mimic the staged introduction of a malaria control intervention. As a consequence, the simulation leaves equilibrium, with host and vector prevalence declining. The *Crash* epoch is allowed to continue until the population has regained equilibrium. At the start of the *Recovery* epoch, we return the changed simulation parameter to its original value, again over thirty days following a logistic transition function. Again, this results in the simulation leaving equilibrium, with the $PfPR$ increasing back to 0.65. As with the *Crash* epoch, the *Recovery* epoch continues until the population regains equilibrium. In summary, the three epoch model allows us to explore both a parasite population decline and rebound.

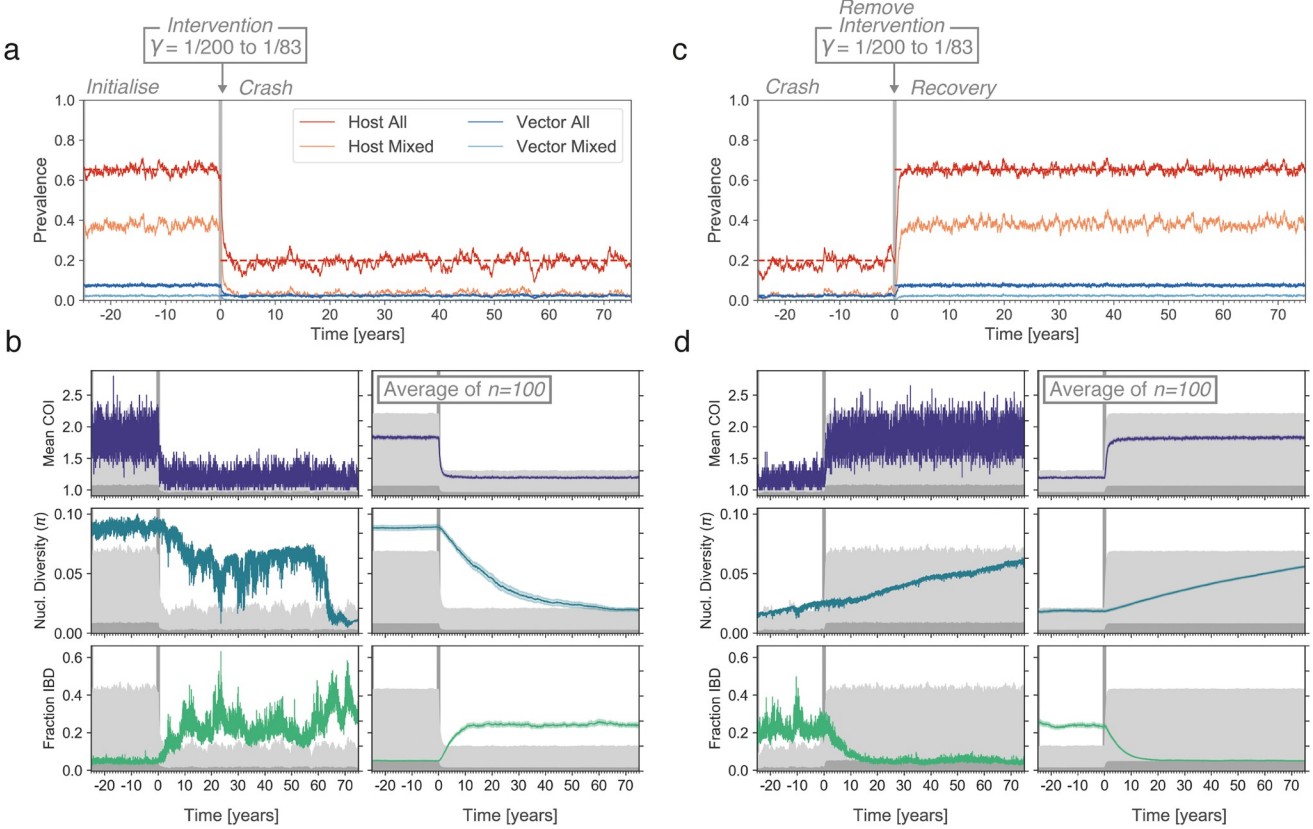

**Fig 4. Responses of genetic diversity statistics to a crash and recovery in parasite prevalence.** (a) An individual `forward-dream` simulation where the number of vectors ($N_v$) is reduced at time zero (x-axis, indicated by grey vertical bar), resulting in a decline of parasite prevalence in hosts from 0.65 to 0.2. (b) Left column, the same simulation as in (a), but showing the response of three genetic diversity statistics (colored lines) to the prevalence change. Light and dark grey areas show host and vector prevalence, as in Fig 2. Right column, mean value of each genetic diversity statistics across 100 independent replicate simulations, with shading showing the 95% confidence interval. (c) Same simulation as in (a) at a later time, where the number of vectors is returned to its original value. Parasite prevalence increases back to 0.65. (d) Same as (b), but corresponding to the recovery shown in (c).

Throughout the three epochs parasite genomes are collected every five days from twenty hosts selected at random, allowing the same suite of genetic diversity statistics discussed above to be followed through time. An immediate observation was that in individual simulations, the trajectories of genetic diversity statistics exhibited considerable noise, tending to fluctuate through time even in the absence of any change to simulation parameters (Fig 4). To better discern the average behaviour of different statistics, we sought to smooth their trajectories by computing a rolling mean. However, consistent with our variance decomposition analysis, we found that the timescale of random fluctuations differs greatly among genetic diversity statistics. Statistics such as the mean COI and the fraction of mixed samples fluctuate rapidly, and variation around their true mean could be substantially reduced (to <20% the original variance) by averaging the genetic data collected over roughly a month (S10 Fig). Yet we found that other statistics, such as nucleotide diversity, are more inertial in nature; they can trend up or down over very long time periods without any change to the underlying simulation parameters. Reducing their initial variance to an equivalent degree required averaging over 10 years worth of genetic data (S10 Fig).

Thus, we took an alternate approach to extract underlying trends: we averaged the trajectories of each statistic across 100 independent, replicate `forward-dream` simulations

  

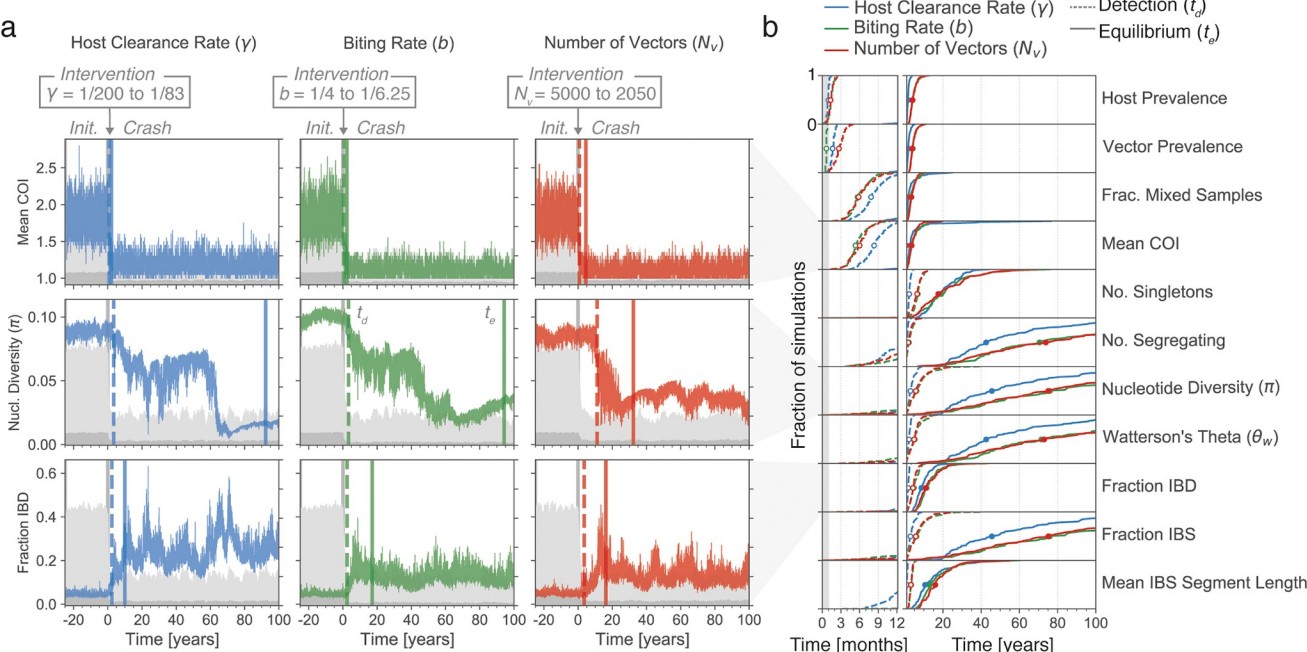

**Fig 5. Detection and equilibrium times of genetic diversity statistics following a crash in parasite prevalence.** (a) Each plot shows the behaviour of a genetic diversity statistic in an individual simulation through a crash in parasite prevalence, induced by: left column, increasing the host clearance rate ($\gamma$); middle column, reducing the vector biting rate ($b$); or right column, reducing the number of vectors. The intervention occurs at time zero (x-axis, grey vertical bar) in all cases. For each plot, the detection time (vertical dashed bar) and equilibrium time (vertical solid bar) of the genetic diversity statistic is indicated. Note that here a single simulation is shown for each intervention type. (b) Empirical cumulative density functions (ECDFs) of the detection and equilibrium times of diversity statistics, created from 100 independent replicate simulations for each intervention type. The y-axis gives the fraction of replicate simulations with a detection (dashed line) or equilibrium (solid line) less than the time indicated on the x-axis. Line color specifies the type of intervention. Open and closed circles give medians for the detection and equilibrium times, respectively. The first year is magnified for clarity.

(Fig 4b and 4d). Several observations emerged. First and consistent with population genetic theory, Tajima's $D$ increases in the period where *PfPR* is declining (indicative of a contracting $N_e$), and falls below zero in the period where *PfPR* is climbing (indicative of an expanding $N_e$) [31] (S11 Fig). Second, there are marked differences in the rate at which different genetic diversity statistics respond to changing prevalence. For example, the COI-related statistics respond faster than IBD or IBS related statistics, which in turn respond faster than nucleotide diversity. Finally, we noted that the rate at which a given genetic diversity statistics responds to a decline in *PfPR* may be different from the rate at which it responds to an increase in *PfPR* (Fig 4).

We developed two metrics to summarize the temporal responses of different genetic diversity statistics to changes in parasite prevalence in our simulations (Fig 5, see also Materials and methods). Within the three epoch simulation framework, we could construct equilibrium distributions for each statistic before and after each prevalence change (i.e. equilibrium distributions for *Initialise*, *Crash*, and *Recovery*). Using these distributions, we computed a "detection time" ($t_d$), which we define as the amount of time, following an intervention, until a given genetic diversity statistic takes on values outside of its pre-intervention equilibrium. In effect, $t_d$ is an estimate of how long it takes to detect that a change in parasite prevalence has occurred by monitoring a given genetic diversity statistic through time. We also computed an "equilibrium time" ($t_e$), which we define as the amount of time until a given genetic diversity statistic reaches its new, post-intervention equilibrium. Note that these metrics were designed to be

informative summaries of the simulations only; it is unlikely they could be deployed in real-world settings.

Fig 5b shows the empirical cumulative density functions for $t_d$ and $t_e$ across all considered genetic diversity statistics, following the host prevalence decline at the beginning of the *Crash* epoch. Consistent with our observations from the averaged trajectories, the mean COI and fraction of mixed samples had both the shortest detection times (median of $\tilde{\mu} = 6$–9 mo., depending on intervention) and equilibrium times ($\tilde{\mu} = 2$–3 yrs). The statistic with the next shortest detection time was the number of segregating sites ($\tilde{\mu} = 1$–2 yrs), however this statistic had a very long equilibrium time ($\tilde{\mu} = 40$–75 yrs). The next two fastest statistics were the fraction IBD and mean IBS segment length. We note these statistics are highly co-linear within our simulations (S4 Fig, Pearson's $R \geq 0.89$), and had both fast detection and equilibrium times (detection $\tilde{\mu} = 1$–4 yrs, equilibrium $\tilde{\mu} = 8$–16 yrs). The least responsive statistic was nucleotide diversity, which had a median detection time of 3–6 years and a median equilibrium time of 45–75 years.

We hypothesised that the fast detection time for the number of segregating sites was a consequence of starting from a relatively high prevalence (65%). In this context, the parasite population carries a large and relatively stable number of segregating sites (CV <10%), and so a significant reduction can be quickly detected. At the same time, IBD related statistics are less sensitive to changes that occur at high prevalence. To explore this further, we repeated the above experiment but declined from an initial prevalence of 30% to a post-intervention prevalence of 10%. Consistent with this hypothesis, the detection time for IBD fraction was faster than number of segregating sites in this context (S12 Fig).

Finally, we sought to understand how the detection and equilibrium times we observed depended on the size of the population under consideration. To this end, we repeated the above experiment but with 400, 700, and 1000 hosts. For statistics related to the genetic genealogy, we found that the equilibrium times grew markedly with the host population size (S13 Fig). For example, the median equilibrium time for nucleotide grew from 28 years for a host population size of 400, to 75 years for a host population size of 1000. Equilibrium times for the mean IBS segment length and Fraction IBD statistics also increased with population size, but to a much lesser degree. Interestingly, the detection times exhibited little dependency on host population size.

In terms of the relative behaviour of the statistics, the $t_d$ and $t_e$ values for the *Recovery* epoch were similar to that of the *Crash* epoch (S14 Fig), with mean COI and the fraction of mixed samples being the fastest statistics, and nucleotide diversity being the slowest. Overall, The median times tended to be longer, in particular for $t_e$. This is likely a consequence of the rate of diversity being re-established (by mutation) being slower than the rate of it being eliminated (by an intervention).

**Seasonality.**   We next aimed to explore whether any measures of genetic diversity were responsive to changes in parasite prevalence driven by seasonality. To this end, we developed a simulation framework where the number of vectors oscillates between a peak reached in the wet season and trough reached in the dry season, with *PfPR* fluctuating between $\sim 0.6$ and $\sim 0.2$ (see Materials and methods). Vectors that die entering the dry season are selected at random, with the dry season lasting 170 days and the wet season lasting 195 days. Consistent with our results from the intervention analysis, we found that the fraction of mixed samples and mean COI showed clear correlations with seasonal change in prevalence ($r^2 = 0.62$ for mean COI, $r^2 = 0.59$ for fraction mixed samples; Fig 6). We found that the mean IBD and IBS segment lengths also exhibited weak correlations with seasonally varying prevalence ($r^2 = 0.12$ and $r^2 = 0.06$, respectively). However, compared to equilibrium patterns, the relationships are in the opposite direction—with an increase in mean segment length at higher prevalence

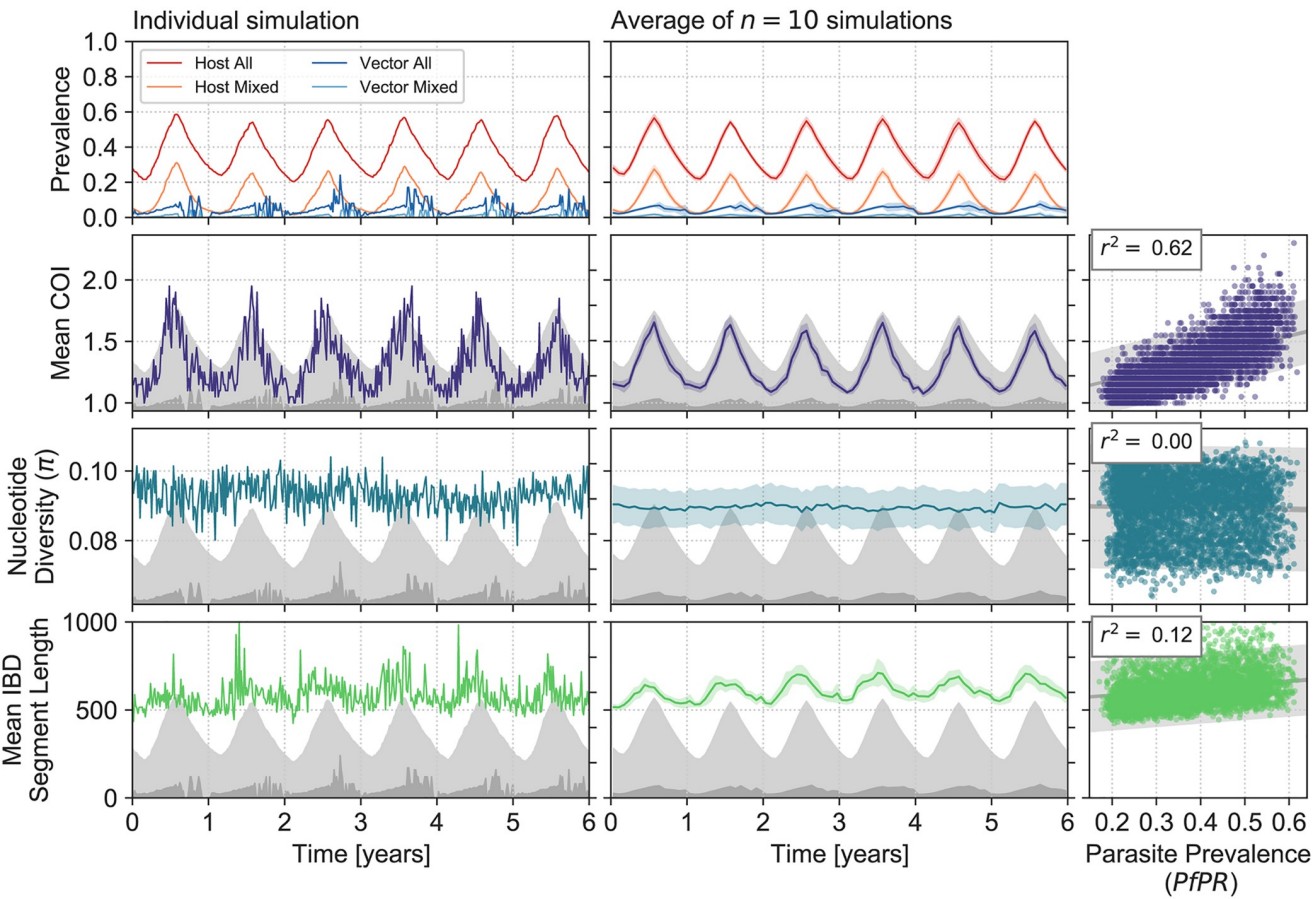

**Fig 6. Responses of genetic diversity statistics to seasonal change in parasite prevalence.** Annual variation in parasite prevalence was induced by varying the number of vectors (top row, see Materials and methods). The behaviour of genetic diversity statistics for an individual simulation is shown in the left column from second row onwards. The mean behaviour of 10 independent replicate simulations is shown at middle, with shaded areas giving the 95% confidence intervals. Scatterplots at right show the relationship between each genetic diversity at parasite prevalence across the six years of seasonal fluctuation. Each point represents a genetic diversity estimate (y-axis) computed from sampling parasite genomes from 20 infected hosts in an individual simulation; parasite prevalence (x-axis) is computed across the entire host population at the same time. Data from all 10 replicate simulations has been aggregated. Variance explained $r^2$ from an ordinary linear regression is indicated at top left.

values. This is likely driven by "epidemic expansion" in the early wet season—with the parasite population expanding faster than it acquires new mutations, resulting in increased IBD [32]. Consistent with this, we observed similar increases in mean IBD and IBS segment length in the first year of the *Recovery* epoch explored in the previous section (S15 Fig).

Notably, none of the other genetic diversity statistics we calculated exhibited correlations with seasonal fluctuation in prevalence.

## Discussion

The collection of parasite genetic data may, over the next few years, become a routine part of malaria surveillance. Yet, deriving the maximum benefit from this data will require an understanding of the relationships between malaria genetic diversity and epidemiology. Such understanding can be guided by modelling approaches, but only if both evolutionary and epidemiological processes are integrated. At present this is rare, as classical models in population genetics are poor approximations of the malaria life cycle, and classical epidemiological models don't incorporate the evolution of parasites. To address this issue, we have combined

the Ross-Macdonald and Moran models into a single framework, which we have implemented as a stochastic simulation called `forward-dream`.

We have used `forward-dream` to investigate the relationships between parasite genetic diversity and parasite prevalence in equilibrium and non-equilibrium settings. We find that many measures of parasite genetic diversity correlate with parasite prevalence at equilibrium. Our findings align with existing empirical data [25, 32–35], and support the idea that the rate of mixed infection (and as a consequence the rate of recombination) is positively correlated with parasite prevalence [36]. Moreover, we find that, for a given human host population size, statistics that reflect the long-term effective population size ($N_e$) of the parasite, such as a nucleotide diversity and the number of segregating sites, also increase with equilibrium prevalence. We also explored the behaviour of these genetic diversity statistics in non-equilibrium settings, most importantly in response to changes in parasite prevalence that mimic malaria control interventions. Other authors have emphasised that the viability of a genomic approach to malaria surveillance will depend on how rapidly signals of epidemiological change become detectable in a reasonably sized sample of parasite genetic data [25]. We find that statistics related to the COI distribution respond most rapidly (on the order of months), whereas other statistics, such as nucleotide diversity, may take decades to respond to a change in parasite prevalence. These pronounced differences in response times are related to the type of events that must occur to change different statistics' values. The COI distribution changes directly in response to individual infection events (on the timescale of the rate of transmission and clearance), whereas other statistics change in response to drift (on the timescale of $N_e$) and/or mutational events (on the timescale of $\mu$).

In addition to understanding the relationships of different genetic diversity statistics with prevalence, it is also important to understand sources of noise and the extent to which they can be controlled. The noise we observe in `forward-dream` has contributions from across the layers of our simulation: for a given set of epidemiological parameters, there will be variation in host prevalence due to the stochastic nature of infection and clearance events; for a given host prevalence, there will be variation in genetic diversity driven by the stochastic nature of drift, mutation and recombination events; and for a given population-level genetic diversity, there will be variation in sequencing data introduced by sampling a subset of all infected hosts. Moreover, on the genetic level, noise can be generated deep within the genealogical tree, related to the timings of coalescent events nearer to the root; and also in the more recent past, driven by more recent coalescent events. The extent to which different statistics are influenced by these sources of noise dictates the value of collecting additional samples, either as part of a cross-sectional survey or longitudinally. For statistics that are influenced by epidemiological noise (such as those derived from the COI distribution) and recent coalescent events (in particular mean length of IBD and IBS segments), increasing sample size can act to average over temporal fluctuations, or reduce sampling variation in a cross-sectional survey. In contrast, collecting additional samples in the present day does little to reduce variation driven by historical events and manifest in statistics like nucleotide diversity or Watterson's Theta ($\theta_w$). For these statistics, relatively few samples are required to produce good estimates, and additional longitudinal sampling in the short-term has little added value. Though not explored here, one way to average over such deep genealogical noise is by assessing a larger proportion of the genome.

Our results also demonstrate how relationships between prevalence and genetic variation are sensitive to the underlying epidemiological process. Specifically, we find that changes to the host clearance rate had a more profound effect on several genetic diversity statistics than changes to either the vector biting rate or density. The statistics exhibiting this behaviour are all related to $\theta = N_e \mu$. As we did not alter the mutation rate across simulations, we expect this

observation is being driven by effects on $N_e$. Where a population's size fluctuates through time, the $N_e$ can be approximated as the harmonic mean of those sizes, and thus is more influenced by periods where the population is small [37]. Similarly, a *P. falciparum* lineage alternates between a large vector population and a small host population, and so one explanation for our observation is that the amount of diversity is impacted more by changes influencing the smaller host population. This result complicates efforts to use genetic variation metrics to compare parasite prevalence across space or time, as it implies that only under certain conditions will changes in prevalence be reflected by changes in genetic variation. Furthermore, it implies that it will be important to assess which modes of prevalence variation are observed more frequently in nature. This assessment can be aided directly by epidemiological data. For example, the empirical observation that the relationship between EIR and prevalence is log-linear suggests that variation in the number of vectors or biting rate may be a dominant epidemiological driver of geographical prevalence variation [38].

`forward-dream` has several limitations, most obviously with respect to its simplicity and scale. With regards to simplicity for example, heterogeneous biting, acquired immunity, and migration are all phenomena that have been proposed to influence the rate of mixed infections [2], yet they are not included in `forward-dream`. We do not explore the effect of selection, though this may be relevant in many contexts, particularly in Southeast Asia where drug resistance is widespread [39]. With regards to scale, we simulate only one thousands hosts, and our genome model is limited to single chromosome of eight thousand sites. As a consequence, some genetic diversity statistics have absolute values that differ from existing empirical observations. Two examples include our estimate of Fraction IBD at low prevalence values, which is higher than observation [40] as a result of the inbreeding that may occur in the small populations we simulate; and our estimate of the number of segregating sites, which is low as a result of our much smaller genome.

Many of these limitations could be addressed with the continued development of `forward-dream`, however, there are at least two salient considerations. The first is that increasing either complexity or scale will likely increase computational costs. Merging the Moran and Ross-Macdonald models resulted in `forward-dream` being more computationally expensive than either, and for most of the individual simulations in this study, run-times were on the order of several hours (S16 Fig). The majority of this was associated with the time required to bring `forward-dream` to equilibrium; a general problem faced by forward-time genetic models (see [41]). Reverse-time simulations harnessing coalescent theory can have greatly accelerated computational times by omitting processes extraneous to the sample of genetic data collected (for example [42]), however the reverse-time formulation of the model described here is yet to be elucidated. In a similar vein, the development of models separating epidemiological and genetic processes are underway [43], and could result in significantly faster simulations. Second, it is important to consider that more complex models typically require more parameters. Most likely these models will be both analytically intractable and statistically non-identifiable, thus making inference about their values impossible without additional and complex field experiments. Indeed, many of the parameters used within the present model have substantial uncertainty and were hard to find in current literature (see S1 Appendix). Community efforts to collate existing knowledge and address key uncertainties through experimental work would greatly benefit the field.

The immediate value of `forward-dream`, as demonstrated here, is as a tool through which relationships between genetics and epidemiology can be explored and experimental and analytical strategies can be evaluated. Moreover, as methods for epidemiological inference from malaria genetic data are developed, `forward-dream` can provide a basis for assessing their expected performance and designing ideal sampling strategies under different

epidemiological scenarios. We do not see `forward-dream` as likely powering such inference methods directly, for example through techniques such as Approximate Bayesian Computation (ABC) (reviewed in [44, 45]), because of the limitations described above. Rather, it will enable the identification of approaches to hypothesis testing and estimation that are insensitive to the specific values of a multitude of epidemiological parameters (for example, assessing relative changes in prevalence by monitoring COI). It is in these ways that `forward-dream`, and other simulations like it, can provide a platform for interpreting the signals within the projected tens of thousands of malaria genomes that will be collected over the next decade, and can help to leverage those signals for malaria surveillance.

## Materials and methods

### Collection of genetic data and computation of summary statistics

For all of the simulations in this manuscript, parasite genomes were collected from randomly selected infected hosts. For each host, we simulated DNA sequencing by taking a subset of all parasite genomes within the host $\bar{h}_k \subseteq h$ (where $h = \{\eta_1, \eta_2, \ldots \eta_{n_h}\}$) such that each genome in $\bar{h}_k$ was different from all others at a minimum of 5% of its sites; the assumption being that genomes more similar than this would not be readily distinguishable by sequencing. The COI of each host is then $k = |\bar{h}_k|$. The fraction of mixed samples and the mean COI is computed directly from the distribution of $k$ across all sequenced hosts. To compute other statistics, we pooled all genomes collected across all hosts. The number of segregating sites, the number of singletons, nucleotide diversity, Watterson's Theta ($\theta_w$), and Tajima's *D* were calculated using `scikit-alllel` (https://scikit-allel.readthedocs.io/en/stable/). We estimated identity-by-descent (IBD) profiles between pairs of parasite genomes by imposing a 2cM length threshold on contiguous segments of identity-by-state (IBS), an approach similar to that used by methods like `GERMLINE` [46].

### Averaging of genetic diversity statistics across independent **forward-dream** simulations

To produce smoothed trajectories of genetic diversity statistics for the intervention analysis (Fig 4b and 4d), we averaged independent replicate `forward-dream` simulations. As `forward-dream` operates in continuous-time, parasite genetic data is never sampled at exactly the same time in independent simulations. Thus, to average simulations, we binned time into 25-day intervals. Finally, across all replicate simulations and for the entire duration of the intervention analysis, genetic diversity statistics computed within each 25-day bin were averaged.

### Computing response time statistics $t_d$ and $t_e$

We created two simple metrics to characterize the temporal response of different genetic diversity to changes in *PfPR*. These metrics were developed specifically in the context of the intervention experiments described in the Results section, and they are computed for individual simulations. The "detection time" ($t_d$) estimates the amount of time before a change in parasite prevalence would be detected, if that change was being monitored for using a given genetic diversity statistic. To compute it, we first construct a distribution for the genetic diversity statistic of interest at equilibrium. When considering a decline in parasite prevalence, this is achieved by recording the genetic diversity statistic's value for 25 years proceeding the *Crash* epoch, during which time the simulation is at equilibrium (at a host prevalence value of 0.65). For an increase, the statistic is recorded for 25 years proceeding the *Recovery* epoch. $t_d$ is then

computed as the first time, after the prevalence chance has occurred, that three consecutive samples have a value for that statistic outside of the quantile interval [$\alpha/2$, 1−$\alpha/2$], with $\alpha$ = 0.01; i.e. the first time when three consecutive samples have a value that would be observed with a probability of less than 1% if the simulation were at equilibrium. Requiring that three consecutive samples (equivalent to approximately two weeks of genetic data) have values outside the interval makes $t_d$ more robust to the high variability observed in individual simulations.

Similarly, the "equilibrium time" ($t_e$) estimates the time until a given genetic diversity statistic regains equilibrium following a host prevalence change. $t_e$ is computed as the first time that six consecutive samples have a value within the inter-quartile range ([$\alpha/2$, 1−$\alpha/2$], with $\alpha$ = 0.5), of the distribution of the statistic at its new equilibrium. Again, requiring six consecutive values within the inter-quartile range makes our estimates of $t_e$ more robust to the high variability of individual simulations; we elected for six rather than only three samples as the criterion of being within the inter-quartile range is weaker than the $t_d$ criterion.

## Parameters for malaria control intervention and seasonality experiments

The complete parameter files (stored as '.ini') used to specify the malaria control intervention and seasonality experiments are available on GitHub within the 'params' directory. All of these experiments began with the same set of parameters listed in Table 1 before individual parameters were changed to either mimic malaria control interventions or induce seasonality. To achieve a parasite prevalence of 0.2 during the *Crash* epoch of malaria control intervention experiments, either the host clearance rate ($\gamma$) was increased to 0.012, the vector biting rate ($b$) was reduced to 0.16, or the number of vectors ($N_v$) was reduced to 2050. In the *Recovery* epoch they were returned to their original values. To achieve an annually varying parasite prevalence in the seasonality experiment, the number of vectors ($N_v$) was oscillated between 10 during a dry season lasting 170 days, and 2800 in the wet season lasting 195 days.

## Supporting information

**S1 Appendix. Additional information on `forward-dream` implementation and parameterisation.**
(PDF)

**S1 Fig. Validating equilibrium host prevalence values in `forward-dream`.** The epidemiological layer of `forward-dream` implements the Ross-Macondald model, where the host prevalence is a function of the rate parameters (see Eq 1). Violinplots summarize the prevalence values observed in `forward-dream` simulations with expected equilibrium prevalence values varying from 0.1 to 0.8 (computed using Eq 1) given on the x-axis. The different equilibrium prevalence values were achieved by varying either the host clearance rate ($\gamma$), the vector biting rate ($b$), or the number of vectors ($N_v$). The variance explained ($r^2$) in an ordinary linear regression is shown at top-left of each plot.
(TIF)

**S2 Fig. Validating intra-host fixation times in `forward-dream`.** (a) The infection of a single host is evolved through time and the within-host alelle frequency of a given site is indicated by the red line. The site fixes around day 125. The experiment is repeated 1000 times (grey lines) and the fraction of infections fixed at a given time is indicated by the blue line. All experiments started with an initial allele frequency of 0.5 and a drift rate of 1/event per day. (b) Distribution of fixation times from (a). The observed mean (64.81 days) is very close to the theoretically expected mean from the Moran model (64.56 days). (c) The experiment in (a) is

repeated but with different initial allele frequencies (x-axis) and three different drift rates (light blue, dark blue, and green line). In all cases, the observed mean fixation times are close to the theoretically expected times. Shading gives 95% confidence intervals for mean estimates.
(TIF)

**S3 Fig. Varying equilibrium prevalence values in `forward-dream`.** The parameter values of `forward-dream` are varied to produce simulations with equilibrium parasite prevalence values varying from 0.1 to 0.8. (a) Varying the number of vectors. Prevalence in hosts indicated in red, vectors in blue. Dots mark parasite prevalence values of 0.1 through 0.8. (b) Varying the vector biting rate $b$. Note $1/b$ gives the average time between successive bites, show in right plot. (c) Varying the host clearance rate ($\gamma$). Note $1/\gamma$ gives the average duration of host infection, shown in right plot.
(TIF)

**S4 Fig. Co-linearity between different genetic diversity statistics in `forward-dream`.** Matrices of Pearson's Correlation Co-efficient ($R$) calculated between all pairs of genetic diversity statistics is shown. In panel (a) host prevalence was tuned to different values between 0.1 and 0.8 by varying the host clearance rate ($\gamma$); (b) by varying the vector biting rate ($b$); or (c) by varying the number of vectors $N_v$. In all cases there is significant co-linearity between different genetic diversity statistics.
(TIF)

**S5 Fig. Equilibrium relationships between parasite prevalence and mixed infection related statistics.** Distributions of mixed infection related genetic diversity statistics (y-axis), plotted for equilibrium parasite prevalence values tuned to between 0.2 and 0.8 (x-axis) in `forward-dream` simulations. Left, middle, and right columns show distributions when parasite prevalence is varied as a function of the host clearance rate ($\gamma$, in blue), vector biting rate ($b$, in green) or number of vectors ($N_v$, in green). Each boxplot contains the result of 30 replicate experiments, where the parasite genomes within 40 randomly selected hosts are collected at every 30 days for 10 years and are used to compute the genetic statistic of interest. The variance explained by ordinary least squares regression is given at top left, and line of best fit and confidence intervals indicated in grey.
(TIF)

**S6 Fig. Equilibrium relationships between parasite prevalence and genetic diversity statistics related to the size and shape of the sample genealogy.** See S5 Fig for details.
(TIF)

**S7 Fig. Equilibrium relationships between parasite prevalence and genetic diversity statistics related to identity-by-descent patterns.** See S5 Fig for details.
(TIF)

**S8 Fig. Exploring the effect of sample size on the noise distributions of genetic diversity statistics.** (a) Shows the influence of sample size on the mean and standard deviation of different genetic diversity statistics. The mean and standard deviation of each genetic diversity statistics across all samples collected (over ten years from thirty replicate simulations) is shown, for three different prevalence levels (indicated by color). Prevalence was varied by changing the number of vectors ($N_v$), and the mean and standard deviation are normalised to their value at a sample size of twenty. Note how increasing sample size reduces the standard deviation of different statistics to varying degrees. (b) Change in $r^2$ for increasing sample sizes. Each row is a different genetic diversity statistic and each column a different sample size (indicated at top by $n$). Individual points within scatter plots represent estimates of the given genetic diversity

statistic from samples of the indicated size, after simulations have reached equilibrium. Shades of green indicate the equilibrium prevalence values of individual simulations. $r^2$ values for all genetic diversity statistics can be found in Fig 3b.
(TIF)

**S9 Fig. Variance decomposition of genetic diversity statistics at equilibrium across a range of prevalence values.** (a) Trajectories of a set of four genetic diversity statistics over a ten year period at equilibrium, for three randomly selected replicate simulations (indicated by color). Each replicate simulation has the same host prevalence of 60%, achieved by tuning the number of vectors $N_v$. Marginal densities for the diversity statistics in each replicate simulation are shown at right. Note how for nucleotide diversity, there is substantial variation between replicate simulations; whereas for other statistics most variation is observed within individual simulations. (b) The fraction of the total variance occurring within- rather than between-individual replicate simulations is shown for all genetic diversity statistics. Each point represents a particular statistic (y-axis), varied epidemiological parameter (color) and equilibrium prevalence value (shade). Analysis is conducted over thirty replicate simulations for each epidemiological parameter and prevalence level.
(TIF)

**S10 Fig. Temporal fluctuations in genetic diversity statistics of different frequencies, without any change in parasite prevalence.** Panel (a) shows the noisy behaviour of three genetic diversity statistics (y-axis) from an individual simulation where parasite prevalence was kept fixed at 0.65 for a 25 year period (x-axis). The trajectory of each statistic was smoothed using a rolling mean, with window sizes varying from 1 day (1 d., purple), which is equivalent to no smoothing, up to 10 years (10 y., yellow). The mean of the statistic during the 25-year window is indicated with the grey horizontal bar. Notice how even with a 10-year window rolling mean, the nucleotide divesity still deviates from its mean value. (b) Across 100 independent replicate simulations, the reduction in variance of each genetic diversity statistic with increasing rolling mean window sizes is shown. The y-axis gives the ratio of the variance for the window size indicated by the x-axis ($Var(X_w)$) divided by the unsmoothed variance in the genetic diversity statistic ($Var(X)$). Increasing with window size of the rolling mean always reduces the variance, but at different rates for different statistics.
(TIF)

**S11 Fig. Average behaviour of Tajima's *D* during a crash and recovery in parasite prevalence.** Colored lines show mean estimate across 100 replicate simulations, shaded area gives 95% confidence intervals. Notice how Tajima's *D* increases during a population contraction and decreases during population growth.
(TIF)

**S12 Fig. Comparing the detection and equilibrium times of genetic diversity statistics following a reduction in the number of vectors from a high prevalence (65%) or low prevalence (30%).** (a) Distributions of detection and equilibrium times are shown for experiments where the prevalence change was from 30% to 10% (Low *PfPR*, blue) or from 65% to 20% (High *PfPR*, red). Each point represents one of one hundred replicate simulations for each experiment. In both cases, prevalence was changed by reducing the number of vectors. Black circle indicates median. (b) Histograms of detection time distributions for Mean COI (blue), Fraction IBD (yellow) and Number of Segregating Sites (green). Vertical dashed lines indicate medians. Notice how the Fraction IBD has a faster median detection time when decline starts at 30%.
(TIF)

**S13 Fig. Comparing detection and equilibrium times for varying host population sizes.** (a) The trajectories of Mean COI, Fraction IBD, and nucleotide diversity ($\pi$) are shown following a reduction in the number of vectors. Colored lines indicate average metric behaviour across 100 replicate simulations, with shaded area indicating standard error of the mean. Host population sizes of 1000 (red), 700 (blue) and 400 (green) are shown. Metrics are scaled to represent percentage of their pre-intervention mean. (b) Distributions of detection and equilibrium times for all statistics across 100 replicate simulations for each host population size. Median is indicated by black circle. Text at right gives median in years and parentheses give rank amongst all metrics for that host population size.
(TIF)

**S14 Fig. Detection and equilibrium times of genetic diversity statistics following a recovery in parasite prevalence.** (a) Each plot shows the behaviour of a genetic diversity statistic in an individual simulation through a recovery of parasite prevalence, induced by: left column, reducing the host clearance rate ($\gamma$); middle column, increasing the vector biting rate ($b$); or right column, increasing the number of vectors. The intervention occurs at time zero (x-axis, grey vertical bar) in all cases. For each plot, the detection time (vertical dashed bar) and equilibrium time (vertical solid bar) of the genetic diversity statistic is indicated. Note that here a single simulation is shown for each intervention type. (b) Empirical cumulative density functions (ECDFs) of the detection and equilibrium times of diversity statistics, created from 100 independent replicate simulations for each intervention type. The y-axis gives the fraction of replicate simulations with a detection (dashed line) or equilibrium (solid line) less than the time indicated on the x-axis. Line color specifies the type of intervention. Open and closed circles give medians for the detection and equilibrium times, respectively. The first year is magnified for clarity.
(TIF)

**S15 Fig. Zoom on response of average IBD track length to increase in prevalence at beginning of *Recovery* epoch.** Focus on IBD and IBS statistics during a four-year window around the beginning of the recovery. Notice how for a change in the number of vectors ($N_v$) there is an increase in average IBS and IBD segment track length at the beginning of the recovery, consistent with epidemic expansion.
(TIF)

**S16 Fig. Peak memory usage and runtime scaling for `forward-dream` simulations.** Left panel shows peak memory usage of individual `forward-dream` simulations brought to equilibrium at different prevalence values. For each prevalence value thirty replicate simulations are plotted; the same simulations analysed in 3. Right panel, same as left but showing run-times of individual simulations (run on 2.6GHz Intel Ivybridge CPUs with 15Gb RAM). Mean run times for each prevalence value are shown at right.
(TIF)

**S1 Table. Genetic diversity statistics computed in `forward-dream`.**
(PDF)

## Acknowledgments

We thank Tim Anderson, Lisa White, Jerome Kelleher, Dan Bridges and Penny Hancock for helpful discussions.

## Author Contributions

**Conceptualization:** Jason A. Hendry, Dominic Kwiatkowski, Gil McVean.

**Formal analysis:** Jason A. Hendry.

**Methodology:** Jason A. Hendry, Gil McVean.

**Software:** Jason A. Hendry, Gil McVean.

**Supervision:** Dominic Kwiatkowski, Gil McVean.

**Visualization:** Jason A. Hendry.

**Writing – original draft:** Jason A. Hendry.

**Writing – review & editing:** Dominic Kwiatkowski, Gil McVean.

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
