## [Decision Letter · Decision Letter 0]

30 Oct 2020

Dear Dr Hendry,

Thank you very much for submitting your manuscript "Elucidating relationships between P.falciparum prevalence and measures of genetic diversity with a combined genetic-epidemiological model of malaria" for consideration at PLOS Computational Biology.

As with all papers reviewed by the journal, your manuscript was reviewed by members of the editorial board and by several independent reviewers. In light of the reviews (below this email), we would like to invite the resubmission of a significantly-revised version that takes into account the reviewers' comments.

We cannot make any decision about publication until we have seen the revised manuscript and your response to the reviewers' comments. Your revised manuscript is also likely to be sent to reviewers for further evaluation.

Sincerely,

Alex Perkins

Associate Editor

PLOS Computational Biology

Virginia Pitzer

Deputy Editor

PLOS Computational Biology

Reviewer's Responses to Questions

**Comments to the Authors:**

Reviewer #1: General Comments:

I really enjoyed reading this paper and think it is a welcome addition to the slowly growing number of malaria transmission models looking at understanding parasite genetics. The study is well set out and explores commonly used metrics to explore the dynamics and utility of different metrics. I do have a few major comments I would like to see addressed and then some smaller minor comments.

Major Comments

COI vs Prevalence Relationship

The COI prevalence relationship seems too low (i.e. not high enough COI at higher prevalences). Looking at estimates derived from THE REAL McCOIL (RMCL) we see evidence in Uganda of mean COI equal to ~7, ~5, and ~2.5. These are in three study sites in Uganda in the original THE REAL McCOIL study (see last table in their SI). RMCL has also been used to estimate COI in 5 countries in https://www.nature.com/articles/s41467-020-15779-8 and the mean COI estimated (see Supp Figure 3) also shows mean COI at least above 2. Also there is good literature on MOI by msp2 by prevalence (Fig 3 in https://journals.plos.org/plosone/article?id=10.1371/journal.pone.0164054), which has been updated in https://academic.oup.com/mbe/advance-article/doi/10.1093/molbev/msaa225/5902837. However, I think it would be easy enough to parameterise your model to this relationship.

IBD

Not sure how you are able to work out IBD if you are only tracking loci as 0 and 1. As you note, you are measuring identity by state. I would recommend not calling it IBD as it is incorrect and should be called IBS.

If you want to track IBD though it should be easy enough. Two approaches. Rather than encoding your genomes as 0/1 encode your loci with integers. (you may need larger than int8 for 400 individuals). Run your simulation until equilibrium prevalence and then replace each individual’s strains with integers, i.e. the first infected individual, all loci are encoded as 1, the next infected individual all loci are 2 etc all the way up to n. Then you are able to track ancestry and IBD relative to this time point. If you also store the genetic population at the time of replacing all your loci in this way, you can then use this as a look up table to work out the actual alleles. Then for mutations, when a mutation occurs, assign that compartment as the next integer value (n+1) and store this in your genetic population table (as well as that integer n+1 is a mutant integer and has the same identity as whichever integer the compartment was before the mutation event). This should then allow you to work out true IBD.

Alternatively, define a population level allele frequency for your 1000 loci. Then draw from those frequencies when seeding your population. Then you could estimate IBD using something like hmmIBD or deploidIBD and report that.

As it is currently, however, I do not agree with it being referred to as IBD and without seeing the comparison between IBD as above and IBS, I do not agree that we know trends in IBD will track with trends in IBS.

Lastly, it is a shame that prevalence less than 20% was not explored as this is where I would expect measures of actual IBD to be most interest (For example, figure 4 in the isorelate methodological paper https://journals.plos.org/plosgenetics/article?id=10.1371/journal.pgen.1007279 shows that it’s only really in SE Asia that you start observing higher IBD sharing). Alternatively, looking at within host IBD comparisons (Figure 4c in the deploidIBD paper https://elifesciences.org/articles/40845#s2) shows the importance of these measures generally at lower transmission intensities. Perhaps worth thinking about if you do look at IBD.

(Also I think it would be fine to say that it is not possible to look at IBD at this stage of fwd-dream, instead reporting IBS statistics and then discuss that this is something to be looked at in future versions).

Superinfection Sub Compartments

The use of sub-compartments to reflect parasites in hosts seems strange or I may have misunderstood something. Why does super infection have to replace previous parasites? I am not aware of any research suggesting that new infections cause old infections to be eliminated. Perhaps it might increase intra-host competition that would drive down the parasite density of older infections more quickly.

Related to sub-compartments. Why do you choose to always replace probabilistically half the genotypes in a superinfection? Should this not depend on the level of cotransmission and the genetic diversity of the incoming sporozoites? As well as perhaps the age/parasite density of present infections? For example, if the vector causing the infection was infected by an individual with a COI of 1, then only one genotype would be introduced but this one genotype being introduced could be causing a large reduction in COI to occur by replacing multiple genetic variants in an infected host that could have had previously a high COI. I wonder if this is one of the reasons for the low COI observed from the model as you are making the assumption that previous strains are lost upon a superinfection event.

Timescales for metrics changing and crash dynamics

The discussion of timescales for metrics changing in response to changing prevalence is a really nice way of judging their utility. However, the reported time for metrics to change is somewhat dependent on the effective population size. You have quite a small population size and so I agree COI and other coinfection metrics will change quickly and relatedness metrics will change more slowly, the specific speed of this is really dependent on the effective population size. It would be good to redo this but with a larger and smaller population size (though may need to be 2 larger population sizes to make sure you can look at the same prevalence range).

The crash dynamics are quite strong, i.e. a large reduction in a short time span. This could be similar to maybe a period of an effective mass drug administration campaign. However, I would be interested to see the variance change in the metrics in response to a more gradual intervention campaign (similar to scale up of bed nets etc.) whereby we see prevalence change by say 30% absolute over 10 years? How useful are each metric in this more subtle change in transmission intensity and how reliably would these metrics be able to detect these changes?

Other Comments

Throughout C.O.I. has been used as opposed to COI which is more common in the literature for complexity of infection. Would recommend replacing in order to align with the literature.

In the introduction you mention the model developed for the Daniels et al Senegal study. I agree this is a bespoke model developed for a specific application and is not currently suitable in its current form for looking at other transmission dynamics. However, very recently a new transmission model for simulating malaria genetics was published - https://academic.oup.com/mbe/advance-article/doi/10.1093/molbev/msaa225/5902837. This appears to address most of the requirements you note as necessary of an integrative genetic-epidemiological model. It would be good to reference this and also to then state the improvements/differences that forward-dream makes.

Line 61: Replace ‘so-called “compartment-based”’ with simply ‘compartment-based’

Drift model could also be cause of low COI. Perhaps this is also another way to parameterise the COI distribution by reducing the frequency of drift events. Also on the parameter estimate chosen for drift events, there may be some data you could use to estimate this perhaps. There are studies where infected individuals have had their parasites sequenced at various time points throughout an infection, often to test for parasite clearance vs reinfection events. I wonder if this information could be used to estimate observed COI in an individual over time, which could help identify drift event frequencies (in my head the time taken for drift events to remove a parasite lineage from a mixed infection could be the same as the time taken for a newly infected individual with a COI equal to 2 to be detected as a monoclonal).

Fig3b where is the biting rate variation for tajimas and no. singletons?

I really enjoyed the discussion in the Supplementary on the “Probability a given parasite genome passes through the host-to-vector (ph) or vector-to-host bottleneck (pv)” about the methodological implications of bottlenecks at the vector-to-host bottleneck. I agree that with high sporozoite counts that it is likely that the majority of genetic diversity in the mosquito will be passed on. Might be worth though highlighting that if the SMFA results of ~5 sporozoites being passed on and defined by a geometric distribution is the correct distribution, then actually a 20% bottleneck will have substantial implications on diversity passed on as we would expect over 50% (cumulative density underneath geometric distribution with mean of 5 after removing zero counts as we are assuming infection is occurring) of infections to have between 1 and 5 sporozoites.

The parameterisation of prevalence is conducted using biting rate, vector abundance and clearance rate. I would think that vector abundance and clearance rate are the more realistic two rather than biting rate. However, these are only explored in parallel. When vector abundance increases and there is increased immunity in the population this then leads to an increase in clearance rate. Not sure it will change the results but would be interested to see how the dynamics in Figure 3b would look when both of these are changed at the same time.

As an aside, it would be interesting to see how forward-dream could be sped up and population size increased. Having gone through the code the first thing that springs to mind re memory is that you have your full human and vector model state as one 3 dimensional array, a lot of which will not be being used when individuals are susceptible. Would be great to see if this could be improved for future versions so that spatial dynamics etc could be explored between multiple populations etc.

Reviewer #2: The manuscript is an attempt by the authors to fill an important gap in the malaria field – translating epidemiologic setting to expected genomic data by linking an epidemiologic and genomic model, the latter including within host as well as transmission elements. A useful model in the space would be quite welcome and have numerous immediate practical applications. The model set up and outputs are clearly described and the manuscript is overall well written, particularly given the potential layers of complexity. While all models are abstractions of reality, they need to at least recover the main features of key relationships have any utility. Unfortunately, I am not convinced that the authors have surpassed this bar, limiting my enthusiasm for what is otherwise a well-performed piece of work.

Major comments:

1) The stated goal of fwd-dream is to provide a basis for assessing the expected performance of methods for inferring transmission intensity from genetic data, and designing sampling strategies for different scenarios (paraphrased from lines 464-7 at the end of the discussion). To have any confidence in utility towards these goals, we would ideally see some model calibration and at a bare minimum at least some very basic “sanity checking” of model outputs with available empiric data on relationships between key transmission parameters and genetic outputs. Unless I have missed it, I do not see any attempt to do this in the manuscript and in fact some of the results presented appear to be red flags. In particular, the percentage of mixed infections and COI (both metrics which are widely, if not systematically, available in the literature) appear to be gross underestimates of what is empirically observed for a given level of parasite prevalence. This first becomes apparent in figure 2, where at 40% parasite prevalence we see only a ~5% prevalence of mixed infections which is much lower available empiric data I am familiar with. In figure 3 (and associated supplemental data) we see COI increase to a maximum of ~1.5 at 80% parasite prevalence, in which setting one might expect much higher COI in most of the population even just based on first principles i.e. nearly everyone is infected, most people are semi-immune and as such have long, untreated infections with considerable superinfection and coinfection. Because these relationships are not believable, it is difficult to know what to make of the other ones, or to be convinced of any of the conclusions regarding the different epidemiologic parameters having differential effects on genetic outputs. The result that metrics of within host diversity would change faster than population level metrics is probably true but does not seem an open question to me given that infections last on the order of one year and most population genetic parameters would be expected to change at a much slower rate.

It is difficult to say if this is a fundamental flaw of the relatively simple model structure or a function of the specific parameters and limited population size/number of years for “burn-in” evaluated (which may derive from the amount of time needed to run the model). Regarding model structure, potential problems include not incorporating immunity, treatment, or age structure, heterogeneity, a limited and fixed number of within host compartments, and if I’m reading things correctly clearing all infections simultaneously as opposed to having independent clearance times for different parasites. Re: run-time parameters, authors could consider a introducing a larger number of diverse parasites at the start instead of 10 identical parasites, allowing for more than 10 within host compartments (which seems like a good number but not if you are replacing half of them with every superinfection and trying to incorporate drift) a much larger host population size (400 seems quite low), running for hundreds or thousands of years instead of 25, etc.

2) Another major limitation of the current model is that it only “works” at transmission levels consistent with 20% or greater parasite prevalence, and many of the metrics of interest to the community, e.g. those based on IBD, may only be expected to show relevant signal in areas of lower transmission.

Minor comments:

3) The authors discuss a metric of IBD throughout the manuscript but I believe what they have actually calculated is IBS. This should be called IBS unless it is actually IBD being calculated.

4) Some terms such as PR and EIR should be properly defined for clarity.

5) Some discussion of how the model could be fitted to real data and used to infer parameters, if this is the goal, should be included.

6) The authors should consider justification of the simplicity of the transmission model, given the considerations above.

Reviewer #3: Hendry et al. tackle a computational and theoretically difficult problem that has rarely been thoroughly addressed before. They aim to link a population-genetic model summarizing within-host genetic variation with a Ross-MacDonald epidemiology/ecological model. Notably, they use the model to study the impact of potential ecology and interventions on common genetic summary statistics. The goal of the model is not directly inference, but is instead an important contribution to our mechanistic and qualitative understanding of a complicated system.

Additionally, one of the most difficult and vague parts of related models is where the parameter values come from. The authors put together a thorough and easy to read summary with many citations and tables in the supplement, which itself is a useful contribution for other papers.

Major comments

1. I appreciate model-based work as a means to understand general qualitative principles, but a main argument the authors make (especially in the introduction) is the need for inference and application to data. While data analysis is beyond the scope of the paper, it would be really beneficial to have a deeper discussion on the types of data, sampling schemes, inference methods, etc to make the model useful to empirically-driven investigators.

2. how does the population size of 400, 100 replicate simulations, or sample of 20 individuals influence results? It seems like a computational decision rather than a biological one? The small sample size is particularly confusing given the high noise. While it is informative to try mimic sampling in data, it would be nice to compare the summary statistics calculated on the full 400 indvs to see where the stochasticity is coming from. Indeed, the authors close the discussion suggesting the tool be used when designing experiments—one of the biggest questions I could foresee the simulations helping with is choosing sample size to sequence under a limited budget, yet no attention is paid to sampling. For example, in Fig 3A, how much of the variation comes from the 30 replicate simulations and how much comes from subsamples of only 20 individuals?

3. Figure 3 shows the equilibrium behavior as a function of parameters that are known to vary or humans can directly influence. However, it is also important to understand how parameter choice for other parameters influences model outcomes and if for example error in one of those produce a signal of similar strength as intervention/biologically varying parameters. Said another way, how much does it matter if parameters such as recombination rate are inaccurate?

Minor comments

•The approach used to study equilibrium behavior is interesting, namely matching the parasite prevalence rather than perturbations in the parameters. I agree this in some ways makes comparisons more equal and interpretable, but it seems important to know the parameter values (perhaps as supplementary information) that are required to produce such parasite prevalences. Are they reasonable?

•Brief summary of the recombination rather than citation of their previous paper would help.

•A number of the intervention timings seem quite dependent on parameters chosen. This would be worth discussing

•DNA sequencing seems to be simulated in a way that only decreases diversity/increases IBD by not differentiating similar strains, but doesn’t account for processes in the other direction such as sequencing error. Is this because with the 5% cutoff similarity, most error would not produce a sequence called as a new strain? How much of an effect on summary statistics does the initial sequencing process make?

**Have all data underlying the figures and results presented in the manuscript been provided?**

Reviewer #1: Yes

Reviewer #2: Yes

Reviewer #3: None

PLOS authors have the option to publish the peer review history of their article (what does this mean?). If published, this will include your full peer review and any attached files.

Reviewer #1: No

Reviewer #2: No

Reviewer #3: No
---

## [Decision Letter · Decision Letter 1]

10 May 2021

Dear Dr Hendry,

Thank you very much for submitting your manuscript "Elucidating relationships between P.falciparum prevalence and measures of genetic diversity with a combined genetic-epidemiological model of malaria" for consideration at PLOS Computational Biology. As with all papers reviewed by the journal, your manuscript was reviewed by members of the editorial board and by several independent reviewers. The reviewers appreciated the attention to an important topic. Based on the reviews, we are likely to accept this manuscript for publication, providing that you modify the manuscript according to the review recommendations.

Sincerely,

Alex Perkins

Associate Editor

PLOS Computational Biology

Virginia Pitzer

Deputy Editor-in-Chief

PLOS Computational Biology

[LINK]

Reviewer's Responses to Questions

**Comments to the Authors:**

Reviewer #1: Thank you to the authors for their revised manuscript. It is much improved and encouraging to see the speed increase. I still have some major concerns though about how accurate the model is in terms of replicating malaria epidemiology and consequently some of the genetic relationships observed. I understand the model tries to be simple (no heterogeneity/immunity) and is still useful for detailing the mean behaviour. However, as a result it is important that the model reflects current evidence and the literature in terms of replication the observed relationships between malaria prevalence, EIR, COI and IBD.

The following are major comments that need to be adequately addressed.

1. The COI vs prevalence relationship has been improved but is still not sufficiently high. The author's response rebutting the validity of the RMCL at accurately estimating COI at high COIs is not to me fair. The authors claim it struggles greater than 4, but that is for COIL - the performance of RMCL is notably more capable at resolving COIs >4, but I will agree that it will struggle at accurately determining say COI > 8. The comments about the prior are not relevant - a flat prior is used because although we know COI is very unlikely to be as high as 25, we know that they can be high (because biting heterogeneity is very overdispersed) and so the flat prior is simplest and unlikely to be altering the inferred COI because it is by definition an uninformative prior. And although RMCL does not have the power to distinguish well between high COIs, it is not acceptable to use this as an argument to have parameterised the COI vs prevalence relationship to have the mean COI still only equal to 2 at 80% prevalence. For example, msp2 MOI literature I referenced in my initial review shows this. Also your comment about DEploidIBD showing lower COIs is because DEploidIBD fails with COIs above 4. For evidence of this, see the recent single cell analysis from Malawi here https://www.sciencedirect.com/science/article/pii/S1931312819306304. In fact, RMCL run on these same samples actually gives higher COIs than DEploidIBD. The relationship between prevalence and COI should still be higher and should scale exponentially given the relationship between EIR and prevalence.

2. EIR outputs. Please could EIR vs prevalence relationships be shown. The relationship between EIR and prevalence is well studied and should be approximately log-linear (https://malariajournal.biomedcentral.com/articles/10.1186/s12936-015-0864-3). I don't think this will be true for a number of the simulations you have done in which prevalence is achieved by altering the clearance rate and keeping the biting rate low. I want to see this relationship as although you have shown that you can recreate different prevalence settings by changing one of three parameters, we have good evidence to know that the biting rate of mosquitoes is unlikely to be as low as 1 feed per 6 days and is closer to 2-3 days climate depending. I think if you constrain this parameter you should be able to identify a set (or range of sets) of the other two that give a suitable EIR vs prevalence relationship as well as a suitable COI vs prevalence relationship (if you don't want to believe the RMCL one then the msp2 literature).

3. Could you plot the relationship between mean pairwise IBD fraction of samples within host against the proportion of mixed infections. I.e. what does it look like in comparison to Figure 4c in the DeploidIBD paper - https://elifesciences.org/articles/40845. The population level pairwise IBD seems very high. At 20% prevalence the Fraction IBD seems close to 0.25, which would mean that on average each parasite is a half sibling of a randomly chosen parasite in the population. That cannot be correct. Have I misunderstood? Is Fraction IBD the fraction of the genomes in mixed infections that are IBD? If it is the population level mean IBD, then this needs to be again addressed to better match relationships observed for IBD. There are a number of studies that have used hmmIBD to estimate population level IBD which would be a starting point - collecting these and plotting the relationship between these and malaria prevalence and compare your relationship (which I thinks shows that fraction IBD is too high).

---

Thank you for the changes made so far and it is encouraging to see the speed gains that have been made so far. Also thank you for keeping your code and analysis notebooks public - has made it much easier to understand what is going on. I look forward to reviewing the manuscript again with these comments addressed.

Reviewer #2: The authors have gone to some length to improve the model performance, analysis, and interpretation. The manuscript is improved and I have no remaining major comments. It is still difficult to know to what degree the method will be useful in exploring any quantitative relationships as opposed to broad, qualitative ones, but the authors defer more quantitative evaluation / calibration for future work. The authors now state directly in the discussion section that the goal of the model at least at present is not e.g. for ABC type estimation but for broad stroke evaluation of relationships, and they have done a sufficient job to justify that use.

If the authors do embark on more quantitative analysis in the future I still feel somewhat comfortable in stating that available empiric data in the literature suggest that the relationship with COI is substantially underestimated with the current model/parameters, hindering potential inference. The authors state in response to Reviewer 1 that the majority of estimates have been using The Real McCOIL and suggest there may be issues with overestimation with this method, but this is a recent approach based on availability of SNP data and for many years the most common way of evaluating within-host diversity (and likely representing the majority of existing literature) has been through genotyping of a limited number of polymorphic markers (initially via length polymorphism of antigens and microsatellites and more recently via amplicon sequencing). These methods may still over or underestimate COI, but most available data including more recent amplicon sequencing data suggest much higher COI. A few references on quick search:

Recent data using amp seq, see supplemental fig 3a and 3b: https://pubmed.ncbi.nlm.nih.gov/31819062/

Using microsats, mean 3.4: https://pubmed.ncbi.nlm.nih.gov/32246127/

Table 4 here has some refs: https://pubmed.ncbi.nlm.nih.gov/33172455/

Table S1 here has some refs: https://pubmed.ncbi.nlm.nih.gov/27711149/

Reviewer #3: The authors have spent considerable time and thought putting together a reasonable and thorough response to reviewer requests. I agree with them that their model, while (as always) an imperfect representation of the real world, is a big improvement to the current available models. More detailed models are susceptible to many assumptions based on processes and parameters we have poor estimates for. They also demonstrate biologically-relevant information that can be learned from their model.

**Have the authors made all data and (if applicable) computational code underlying the findings in their manuscript fully available?**

Reviewer #1: Yes

Reviewer #2: Yes

Reviewer #3: None

PLOS authors have the option to publish the peer review history of their article (what does this mean?). If published, this will include your full peer review and any attached files.

Reviewer #1: No

Reviewer #2: No

Reviewer #3: **Yes: **Amy Goldberg

Figure Files:

Data Requirements:

Reproducibility:

References:

---

## [Decision Letter · Decision Letter 2]

19 Jul 2021

Dear Dr Hendry,

We are pleased to inform you that your manuscript 'Elucidating relationships between P.falciparum prevalence and measures of genetic diversity with a combined genetic-epidemiological model of malaria' has been provisionally accepted for publication in PLOS Computational Biology.

Best regards,

Alex Perkins

Associate Editor

PLOS Computational Biology

Virginia Pitzer

Deputy Editor-in-Chief

PLOS Computational Biology

Reviewer's Responses to Questions

**Comments to the Authors:**

Reviewer #1: I feel the authors have done sufficient work to demonstrate that the model developed is a helpful tool. Personally, I feel it would be more helpful for the authors to present model defaults that better capture certain relationships observed but I am also more than happy for that to remain as an academic differing of opinions and it is not one that should prevent this paper being published. I am very happy that they have taken the time to explain that the tool is more useful for exploring relationships in a qualitative fashion rather than being used to provide quantitative predictions of specific metrics and detailing that some of this results from the population sizes that are explored by the model. As such I am satisfied with their revisions. Once again thank you for making all code and analysis scripts open and easy to review.

**Have the authors made all data and (if applicable) computational code underlying the findings in their manuscript fully available?**

Reviewer #1: Yes

PLOS authors have the option to publish the peer review history of their article (what does this mean?). If published, this will include your full peer review and any attached files.

Reviewer #1: No

---

## [Editor Report · Acceptance letter]

16 Aug 2021

PCOMPBIOL-D-20-01626R2 

Elucidating relationships between *P.falciparum* prevalence and measures of genetic diversity with a combined genetic-epidemiological model of malaria

Dear Dr Hendry,

I am pleased to inform you that your manuscript has been formally accepted for publication in PLOS Computational Biology. Your manuscript is now with our production department and you will be notified of the publication date in due course.

With kind regards,

Olena Szabo
